# SGD with Adaptive Preconditioning: Unified Analysis and Momentum Acceleration

**Dmitry Kovalev**
Yandex Research
dakovalev1@gmail.com

## Abstract

In this paper, we revisit stochastic gradient descent (SGD) with AdaGrad-type preconditioning. Our contributions are twofold. First, we develop a unified convergence analysis of SGD with adaptive preconditioning under anisotropic or matrix smoothness and noise assumptions. This allows us to recover state-of-the-art convergence results for several popular adaptive gradient methods, including AdaGrad-Norm, AdaGrad, and ASGO/One-sided Shampoo. In addition, we establish the fundamental connection between two recently proposed algorithms, Scion and DASGO, and provide the first theoretical guarantees for the latter. Second, we show that the convergence of methods like AdaGrad and DASGO can be provably accelerated beyond the best-known rates using Nesterov momentum. Consequently, we obtain the first theoretical justification that AdaGrad-type algorithms can simultaneously benefit from both diagonal preconditioning and momentum, which may provide an ultimate explanation for the practical efficiency of Adam.

## 1 Introduction

The optimization community has shown strong interest in adaptive stochastic gradient optimization methods over recent years (Duchi et al., 2011; Tieleman, 2012; Kingma & Ba, 2014; Gupta et al., 2018; Reddi et al., 2019) due to their applications in deep learning (LeCun et al., 2015). This research direction has notably led to the development of Adam (Kingma & Ba, 2014) and AdamW (Loshchilov & Hutter, 2017), algorithms with remarkable performance in training deep neural networks. Unfortunately, despite almost a decade of research, these algorithms continue to be the preferred choice for most deep learning tasks, particularly in the training of large language models (Achiam et al., 2023; Liu et al., 2024a; Grattafiori et al., 2024; Anil et al., 2023). The lack of worthy contenders to Adam and AdamW may be attributed to insufficient theoretical understanding of adaptive optimization algorithms. Therefore, the primary objective of this paper is to enhance the theoretical comprehension of this research area. Formally speaking, we consider the following optimization problem:

$$\min_{x \in \mathcal{X}} f(x), \tag{1}$$

where $\mathcal{X}$ is a finite-dimensional Euclidean space, and $f(x) \colon \mathcal{X} \to \mathbb{R}$ is a continuous convex[1] objective function. We assume that problem (1) has a solution $x^* \in \mathcal{X}$.

### 1.1 Baseline Algorithm: AdaGrad

The starting point for the development of Adam and AdamW was the gradient descent (GD) with the AdaGrad-Norm stepsizes (Streeter & McMahan, 2010). Given the parameter $\eta > 0$ and the past gradients $g_i \in \partial f(x_i)$ for $i = 0, \ldots, k$, this algorithm performs the following update:

$$x_{k+1} = x_k - \eta_k g_k, \quad \text{where} \quad \eta_k = \frac{\eta}{\sqrt{\sum_{i=0}^{k} \|g_i\|^2}}. \tag{2}$$

It is well known that AdaGrad-Norm can achieve the convergence rate $\mathcal{O}(1/K)$ of GD with fixed stepsizes for smooth functions with Lipschitz-continuous gradients and the rate $\mathcal{O}(1/\sqrt{K})$ of GD

---

[1]We discuss the justification for using the convexity assumption in Appendix C.

with diminishing step sizes for non-smooth Lipschitz functions or when only stochastic gradients are available (Orabona, 2023; Li & Orabona, 2019; Levy et al., 2018). However, the main benefit of this algorithm is that it can achieve both rates with the single parameter choice $\eta \propto \|x^*\|$. In other words, it can adapt to the level of smoothness and gradient noise of the function $f(x)$, which is called "universality" (Nesterov, 2015). Furthermore, Duchi et al. (2011); McMahan & Streeter (2010) proposed the AdaGrad method, which performs a coordinate-wise variant of the update (2), aiming to exploit the potential sparsity of the gradients $g_k$. Although they provided a limited theoretical justification for the benefits of coordinate-wise updates compared to scalar stepsizes (2), AdaGrad and its modifications, such as RMSProp (Tieleman, 2012) and Adam, have proven to be highly efficient in practice.

## 1.2 Adaptive Gradient Methods with Structured Preconditioning

Motivated by the success of AdaGrad, many adaptive optimization algorithms has been developed that fall into the category of gradient methods with preconditioning. Such algorithms use the update rule of the form

$$x_{k+1} = \arg\min_{x \in \mathcal{X}} \langle g_k, x \rangle + \tfrac{1}{2}\|x - x_k\|^2_{\mathbf{H}_k^{-1}}, \tag{3}$$

where $\mathbf{H}_k \in \mathbb{S}_{++}$ is a symmetric positive definite preconditioning operator $\mathcal{X} \to \mathcal{X}$. Besides AdaGrad, which uses a diagonal preconditioning matrix, notable examples of such algorithms include Shampoo (Gupta et al., 2018) and its theoretically streamlined variants: One-sided Shampoo (Xie et al., 2025) and ASGO (An et al., 2025). Motivated by the structure of neural networks, these algorithms are specifically designed for optimizing the function $f(X) \colon \mathbb{R}^{m \times n} \to \mathbb{R}$ of an $m \times n$ matrix argument and use preconditioners that respect the function's structure. In particular, One-sided Shampoo and ASGO use the preconditioner $\mathbf{H}_k \colon G \mapsto (\sum_{i=0}^{k} G_i G_i^\top)^{-1/2} G$, where $G \in \mathbb{R}^{m \times n}$ and $G_i \in \partial f(X_i)$. Overall, the practical performance of Shampoo and its Adam-like modification, SOAP (Vyas et al., 2024), is comparable to that of Adam and sometimes exceeds it.

Here, we come to the following issue: every time an adaptive preconditioned gradient method is developed, one has to provide a separate convergence proof, even though the update rules in such algorithms, as well as the convergence proofs, often have a similar structure. Consequently, we arrive to the following question:

> **Q1.** *Can we develop a unified convergence analysis that would cover most existing adaptive preconditioned gradient methods, including AdaGrad, Shampoo, ASGO, etc.?*

A positive answer to this question was partially provided by the unified approach of Gupta et al. (2017), who showed that the preconditioner operator $\mathbf{H}_k$ can be defined as a solution to a certain optimization problem over a linear subspace of self-adjoint operators $\mathcal{H} \subset \mathbb{S}$. For instance, the update rule for AdaGrad-Norm and AdaGrad can be obtained by choosing $\mathcal{H}$ to be the space of multiples of the identity and the space of diagonal operators, respectively. Unfortunately, the unified approach of Gupta et al. (2017) has major flaws: it still requires separate convergence proofs for different algorithms, provides convergence guarantees only for non-smooth functions, and offers no explanation for the benefits of using general preconditioning operators.

## 1.3 Matrix Smoothness and Acceleration

**Matrix smoothness.** In an attempt to find a theoretical justification for the success of adaptive preconditioned gradient methods, a considerable amount of recent research has focused on developing theoretical analyses of such methods under the assumption that the function smoothness, as well as the gradient noise level, is measured in terms of the weighted Euclidean norm $\|\cdot\|_{\mathbf{B}}$, where $\mathbf{B} \in \mathbb{S}_{++}$ is a self-adjoint positive definite operator. For instance, Liu et al. (2024b); Jiang et al. (2024) provided an analysis of AdaGrad under anisotropic smoothness, i.e., in the case of the diagonal operator $\mathbf{B} \colon x \mapsto \boldsymbol{b} \odot x$, where $\boldsymbol{b}, x \in \mathbb{R}^d$. When the vector $\boldsymbol{b}$ is sparse, they managed to prove substantially better theoretical convergence guarantees for AdaGrad compared to AdaGrad-Norm, thus obtaining theoretical justification for the practical benefits of diagonal preconditioning. Similarly, An et al. (2025); Xie et al. (2025) considered the matrix smoothness, i.e., the case where the operator $\mathbf{B} \colon X \mapsto BX$, where the matrix $B \in \mathbb{R}^{m \times m}$ is symmetric and positive definite, and $X \in \mathbb{R}^{m \times n}$. This allowed them to theoretically justify the practical success of Shampoo-like al-

gorithms. However, Question 1 discussed above is relevant here: a separate convergence proof is required for each algorithm, even though they share many similarities.

**Momentum acceleration.** Besides diagonal preconditioning, momentum is another key component that contributes to the efficiency of Adam. It is well-known that Nesterov momentum (Nesterov, 1983) can accelerate the convergence of GD for smooth convex (Nesterov, 2013) and convex-like (Hinder et al., 2020) functions up to the rate $\mathcal{O}(1/T^2)$. Consequently, there is an array of works that aim to establish theoretical guarantees for AdaGrad-type methods with Nesterov acceleration, including the works of Levy et al. (2018); Cutkosky (2019); Kavis et al. (2019); Rodomanov et al. (2024); Kreisler et al. (2024). However, to the best of our knowledge, all such algorithms achieve accelerated theoretical convergence rates only for scalar stepsizes. Therefore, another natural question appears:

> **Q2.** *Can we design an adaptive preconditioned gradient method that provably benefits from both diagonal AdaGrad-type preconditioning and momentum?*

To the best of our knowledge, the only attempt to answer this question was made by Trifonov et al. (2025). However, they made additional unrealistic assumptions about the dynamics of the preconditioning operator and considered only a smooth and strongly convex, non-stochastic setting. Their theoretical results provided a highly limited explanation of the benefits of preconditioning, including a lack of adaptation to stochasticity and matrix/anisotropic Hölder smoothness.

### 1.4 CONTRIBUTIONS AND RELATED WORK

In this paper we give positive answers to Questions 1 and 2 and provide the following contributions:

(i) We develop a unified analysis framework for adaptive preconditioned stochastic gradient methods under the matrix Hölder smoothness and bounded variance. Using this framework, in Section 3, we provide a single convergence proof that is applicable to most existing AdaGrad-type algorithms, recovering the state-of-the-art convergence guarantees for AdaGrad-Norm, AdaGrad, and ASGO/One-sided Shampoo. Moreover, we establish convergence guarantees for DASGO, a computationally efficient variant of ASGO proposed by An et al. (2025), and find its fundamental connection with the recently proposed Scion method by Pethick et al. (2025).

(ii) We develop a novel unified analysis of adaptive preconditioned stochastic gradient methods with Nesterov acceleration under the additional assumption that the smoothness and noise operators,[2] $\mathbf{L}$ and $\boldsymbol{\Sigma}$, commute with any preconditioner $\mathbf{H}_k$. In particular, in Section 4, we show that the convergence of algorithms with diagonal preconditioning, such as AdaGrad and DASGO, can be significantly improved with no extra assumptions compared to their non-accelerated counterparts. To the best of our knowledge, this is the first theoretical justification that AdaGrad can benefit from both momentum and diagonal preconditioning.

We also provide a discussion of additional related work. First, we discuss the theoretical analysis of the exponential moving average (EMA) in AdaGrad-type algorithms by Défossez et al. (2020). Second, we mention several parameter-free AdaGrad-type algorithms that do not require tuning the parameter $\eta$. Finally, we discuss the concurrent unified analysis of AdaGrad-type algorithms by Xie et al. (2025), which appeared online earlier than our work but suffers from several substantial drawbacks. The details are postponed to Appendix B due to the maximum page limit.

## 2 PRELIMINARIES

### 2.1 UNIFIED PRECONDITIONING FRAMEWORK

In this paper, we use the notation described in Appendix A. The preconditioned gradient method uses the update rule in eq. (3), which requires the preconditioning operator $\mathbf{H}_k \in \mathbb{S}_{++}$. Similar to the approach of Gupta et al. (2017), we restrict the operator $\mathbf{H}_k$ to belong to a certain subspace of self-adjoint operators $\mathcal{H} \subset \mathbb{S}$. As discussed in Section 1.2, we can obtain most existing AdaGrad-type methods by choosing different instances of the space $\mathcal{H}$. However, to develop a single unified

---

[2]Refer to Assumptions 2 and 3 for precise definitions.

convergence proof for these algorithms, we need to impose formal assumptions on the space $\mathcal{H}$. This is done through the following Definition 1 and Assumption 1.

**Definition 1.** *Let $\psi(h)\colon I \to \mathbb{R}$ be a scalar function defined on an arbitrary interval $I \subset \mathbb{R}$. Let $\mathcal{S}_I \subset \mathbb{S}$ be the set of self-adjoint operators, with eigenvalues lying in $I$. The corresponding operator function $\psi(\mathbf{H})\colon \mathcal{S}_I \to \mathbb{S}$ is defined as follows:*

$$\psi(\mathbf{H}) = \sum_i \psi(\lambda_i)\mathbf{P}_i, \tag{4}$$

*where $\mathbf{H} = \sum_i \lambda_i \mathbf{P}_i$ is the eigendecomposition of the operator $\mathbf{H} \in \mathcal{S}_I$, that is, $\lambda_i \in I$ are the eigenvalues of $\mathbf{H}$, and $\mathbf{P}_i \in \mathbb{S}$ are the projection operators onto the corresponding eigenspaces.*

**Assumption 1.** *The space of linear operators $\mathcal{H} \subset \mathbb{S}$ satisfies the following properties:*

**(A1.1)** *The projection onto $\mathcal{H}$ is order preserving, that is, $\mathrm{proj}_{\mathcal{H}}(\mathbf{H}) \in \mathbb{S}_{++}$ for all $\mathbf{H} \in \mathbb{S}_{++}$.*

**(A1.2)** *The space $\mathcal{H}$ is closed under arbitrary operator functions, that is, $\psi(\mathbf{H}) \in \mathcal{H}$ for all $\mathbf{H} \in \mathcal{H}$ and $\psi(h)\colon \mathbb{R} \to \mathbb{R}$.*

Next, according to Gupta et al. (2017), we describe a unified way to define the preconditioning operator $\mathbf{H}_k \in \mathbb{S}_{++}$ based on the choice of the space $\mathcal{H}$. Given the past gradients $g_0, \dots, g_k \in \mathcal{X}$, the preconditioning operator $\mathbf{H}_k$ is defined as a solution to the following optimization problem:

$$\mathbf{H}_k = \underset{\mathbf{H}\in\mathcal{H}\cap\mathbb{S}_{++}}{\arg\min} \langle \mathbf{H}, \mathbf{S}_k \rangle + \langle \mathbf{I}, \phi(\mathbf{H}) \rangle, \quad \text{where} \quad \mathbf{S}_k = \sum_{i=0}^{k} g_i\langle g_i, \cdot\rangle, \tag{5}$$

where $\phi(h)\colon \mathbb{R}_{++} \to \mathbb{R}$ is a strictly convex non-negative potential function. The optimization form of this definition allows the use of the standard tool from online optimization, the Follow-the-Leader/Be-the-Leader (FTL-BTL) lemma (Kalai & Vempala, 2005). It can be summarized in the following inequality:

$$\sum_{i=-1}^{k} l_i(\theta_i) \le \sum_{i=-1}^{k} l_i(\theta_k), \quad \text{where} \quad \theta_k = \arg\min_{\theta\in\Theta}\sum_{i=-1}^{k} l_i(\theta), \quad \text{(FTL-BTL)}$$

where $l_{-1}(\theta), \dots, l_k(\theta)\colon \Theta \to \mathbb{R}$ is an arbitrary sequence of functions defined on a domain $\Theta$.[3] Similar to Gupta et al. (2017), we can use this result to obtain the following Lemma 1, which is one of the key elements in the unified analysis of Adagrad-type algorithms.

**Lemma 1** ($\downarrow$)**.** *The preconditioner $\mathbf{H}_k$ defined in eq. (5) satisfies the following inequality:*

$$\sum_{i=0}^{k} \|g_i\|_{\mathbf{H}_i}^2 \le \langle \mathbf{H}_k, \mathbf{S}_k \rangle + \langle \mathbf{I}, \phi(\mathbf{H}_k) \rangle. \tag{6}$$

The application of Lemma 1 is not limited to a specific choice of the potential function. However, to obtain Adagrad-type preconditioners, we will use the following potential function $\phi(h)$, which is given as follows:

$$\phi(h) = \delta \cdot h + \eta^2/h, \tag{7}$$

where $\delta, \eta > 0$ are positive parameters. Here appears the first key difference from Gupta et al. (2017): using our Assumption 1, we can explicitly compute the preconditioner $\mathbf{H}_k$, as stated by the following Lemma 2.

**Lemma 2** ($\downarrow$)**.** *The auxiliary problem in eq. (5) with the potential function $\phi(h)$ defined in eq. (7) has the following unique solution:*

$$\mathbf{H}_k = \eta \left(\delta\mathbf{I} + \mathrm{proj}_{\mathcal{H}}(\mathbf{S}_k)\right)^{-1/2}. \tag{8}$$

*Moreover, the following operator inequality holds:*

$$\mathbf{H}_{k+1} \preceq \mathbf{H}_k. \tag{9}$$

Overall, the assumptions that we impose on the space of preconditioning operators $\mathcal{H}$ (Properties A1.1 and A1.2 in Assumption 1) are closely related to the notion of a well-structured preconditioner set used by Xie et al. (2025). Consequently, the unified analysis of Xie et al. (2025) shares some similarities with ours but suffers from significant disadvantages discussed in Appendix B.

---

[3]The proof of eq. (FTL-BTL) can be found in Appendix A of Gupta et al. (2017).

Table 1: The linear space $\mathcal{X}$, the space of preconditioning operators $\mathcal{H}$ satisfying Assumption 1, and the (possibly non-Euclidean) norm $\mathcal{R}(\cdot)$ defined in eq. (12) for AdaGrad-Norm (Streeter & McMahan, 2010), AdaGrad (Duchi et al., 2011; McMahan & Streeter, 2010), ASGO/One-sided Shampoo (An et al., 2025; Xie et al., 2025), and DASGO (An et al., 2025).

| Algorithm | $\mathcal{X}$ | $\mathcal{H}$ | $\mathcal{R}(\cdot)$ |
|---|---|---|---|
| AdaGrad-Norm | $\mathbb{R}^d$ | $\{g \mapsto \beta g : \beta \in \mathbb{R}\}$ | $\frac{1}{\sqrt{d}}\|\cdot\|$ |
| AdaGrad | $\mathbb{R}^d$ | $\{g \mapsto \boldsymbol{b} \odot g : \boldsymbol{b} \in \mathbb{R}^d\}$ | $\|\cdot\|_\infty$ |
| ASGO/One-sided Shampoo | $\mathbb{R}^{m \times n}$ | $\{G \mapsto BG : B \in \mathbb{S}^m\}$ | $\frac{1}{\sqrt{n}}\sigma_{\max}(\cdot)$ |
| DASGO | $\mathbb{R}^{m \times n}$ | $\{G \mapsto \operatorname{diag}(\boldsymbol{b})G : \boldsymbol{b} \in \mathbb{R}^m\}$ | $\frac{1}{\sqrt{n}}\|\cdot\|_{2\to\infty}$ |

## 2.2 Assumptions on the Objective Function

In this section, we formalize the assumptions that we impose on the objective function $f(x)$. The following Assumption 2 formalizes the convexity and matrix Hölder smoothness properties of the function $f(x)$. Note that in the smooth case ($\nu = 1$) Assumption 2 matches the definitions used by An et al. (2025); Xie et al. (2025). In the non-smooth case ($\nu = 0$), it is more general compared to the assumption used by An et al. (2025, Corollary 2). Note that Xie et al. (2025) provides no results in the non-smooth case, and neither of the works of An et al. (2025); Xie et al. (2025) provides results in the Hölder smooth case for $0 < \nu < 1$.

**Assumption 2.** *The function $f(x)$ is convex and $(\|\mathbf{L}\|_{\mathrm{tr}}^{\frac{1-\nu}{2}}, \nu)$-Hölder smooth with respect to the norm $\|\cdot\|_{\mathbf{L}}$, where $\nu \in [0, 1]$ and $\mathbf{L} \in \mathcal{H} \cap \mathbb{S}_{++}$. That is, for all $x_1, x_2 \in \mathcal{X}$ and $\nabla f(x_1) \in \partial f(x_1)$, the following inequalities hold:*

$$0 \leq f(x_2) - f(x_1) - \langle \nabla f(x_1), x_2 - x_1 \rangle \leq \frac{1}{1+\nu}\|\mathbf{L}\|_{\mathrm{tr}}^{\frac{1-\nu}{2}}\|x_2 - x_1\|_{\mathbf{L}}^{1+\nu}. \tag{10}$$

Additionally, using the matrix Hölder smoothness property in Assumption 2, we establish the following Lemma 3, which will be further used in our convergence analysis.

**Lemma 3** ($\downarrow$)**.** *For all $x \in \mathcal{X}$ and $\nabla f(x) \in \partial f(x)$, the following inequality holds:*

$$\|\nabla f(x)\|_{\mathbf{L}^{-1}}^2 \leq \left(\frac{1+\nu}{\nu}\right)^{\frac{2\nu}{1+\nu}}\|\mathbf{L}\|_{\mathrm{tr}}^{\frac{1-\nu}{1+\nu}}\left(f(x) - f(x^*)\right)^{\frac{2\nu}{1+\nu}}, \tag{11}$$

*where in the case $\nu = 0$, we use the convention $0^0 = 1$.*

The matrix smoothness in Assumption 2 is also closely related to the non-Euclidean smoothness property, which recently received a lot of attention (Bernstein & Newhouse, 2024; Pethick et al., 2025; Kovalev, 2025; Riabinin et al., 2025) due to the practical success of the Muon optimizer (Jordan et al., 2024). Let function $\mathcal{R}(x): \mathcal{X} \to \mathbb{R}_+$ be defined as follows:

$$\mathcal{R}(x) = \|\operatorname{proj}_{\mathcal{H}}(\mathbf{X})\|_{\mathrm{op}}^{1/2}, \quad \text{where } \mathbf{X} = x\langle x, \cdot\rangle. \tag{12}$$

One can verify that the function $\mathcal{R}(x)$ is a norm on the linear space $\mathcal{X}$, as shown in Lemma 4. Besides, Assumption 2 implies that the function $f(x)$ is $(\|L\|_{\mathrm{tr}}, \nu)$-Hölder smooth with respect to this possibly non-Euclidean norm $\mathcal{R}(\cdot)$. That is, the following inequality holds for all $x_1, x_2 \in \mathcal{X}$:

$$f(x_2) - f(x_1) - \langle \nabla f(x_1), x_2 - x_1 \rangle \leq \frac{1}{1+\nu}\|\mathbf{L}\|_{\mathrm{tr}}\left[\mathcal{R}(x_2 - x_1)\right]^{1+\nu}. \tag{13}$$

We provide additional discussion of the connection between Assumption 2 and the non-Euclidean Hölder smoothness in eq. (13) in Section 3.

**Lemma 4** ($\downarrow$)**.** *The function $\mathcal{R}(x)$ defined in eq. (12) is a norm. That is, it is subadditive, absolutely homogeneous, non-negative, and positive definite.*

Additionally, we provide the assumptions on the stochastic gradient noise in the following Assumption 3. These are more general than the assumptions used by both An et al. (2025) and Xie et al.

---

**Algorithm 1** Adaptive SGD with Preconditioning

---

1: **input:** $x_0 \in \mathcal{X}$, $K \in \{1, 2, \dots\}$
2: **for** $k = 0, \dots, K$ **do**
3:      sample $\xi_k \sim \mathcal{D}$
4:      compute $g_k = \nabla f(x_k; \xi_k)$
5:      compute $\mathbf{H}_k \in \mathcal{H} \cap \mathbb{S}_{++}$ using eqs. (5) and (8)
6:      compute $x_{k+1} \in \mathcal{X}$ using eq. (3).
7: **output:** $\overline{x}_K = \frac{1}{K+1} \sum_{k=0}^{K} x_k$

---

(2025). In particular, they assume the ordering $\mathbb{E}_{\xi \sim \mathcal{D}}[n(x; \xi)\langle n(x; \xi), \cdot\rangle] \preceq \mathbf{\Sigma}^2$, which implies Property A3.2, and hence, is more restrictive. Moreover, similar to the connection between Assumption 2 and the non-Euclidean Hölder smoothness (13), one can show that Assumption 3 implies that the variance of the stochastic gradient estimator is bounded with respect to the non-Euclidean dual norm $\mathcal{R}^*(\cdot)$. That is, the following inequality holds for all $x \in \mathcal{X}$:

$$\mathbb{E}_{\xi \sim \mathcal{D}}\left[(\mathcal{R}^*(n(x; \xi)))^2\right] \leq \|\mathbf{\Sigma}\|_{\mathrm{tr}}^2. \tag{14}$$

**Assumption 3.** *There exists a stochastic estimator $\nabla f(x; \xi) = n(x; \xi) + \nabla f(x)$ of the (sub)gradient $\nabla f(x) \in \partial f(x)$ of the objective function $f(x)$, where $n(x; \xi)$ is the noise and $\xi \sim \mathcal{D}$ is a random variable. The noise $n(x; \xi)$ satisfies the following properties:*

**(A3.1)** *Zero mean: $\mathbb{E}_{\xi \sim \mathcal{D}}[n(x; \xi)] = 0$ for all $x \in \mathcal{X}$.*

**(A3.2)** *Bounded variance: $\mathbb{E}_{\xi \sim \mathcal{D}}[\|n(x; \xi)\|_{\mathbf{\Sigma}^{-1}}^2] \leq \|\mathbf{\Sigma}\|_{\mathrm{tr}}$ for all $x \in \mathcal{X}$, where $\mathbf{\Sigma} \in \mathcal{H} \cap \mathbb{S}_{++}$.*

## 3 Unified Analysis of Adaptive SGD with Preconditioning

### 3.1 General Algorithm and its Convergence

Based on the discussion in Section 2.1, we formalize the adaptive stochastic gradient method with preconditioning as Algorithm 1. In this section, we develop the unified convergence analysis of this algorithm. First, we obtain an upper bound on the expected regret $\mathbb{E}[\sum_{k=0}^{K} f(x_k) - f(x^*)]$ in the following Lemma 5. The proof of this lemma, in many ways, relies on the previously obtained Lemmas 1 and 2.

**Lemma 5** ($\downarrow$). *Under the conditions of Theorem 1, the following inequality holds:*

$$\sum_{k=0}^{K} \mathbb{E}[f(x_k) - f(x^*)] \leq \tfrac{3}{2}\mathcal{R}\langle \mathbf{I}, \mathrm{proj}_{\mathcal{H}}(\mathbb{E}[\mathbf{S}_K])^{1/2}\rangle + \tfrac{3}{2}\sqrt{\delta}\mathcal{R}\dim(\mathcal{X}). \tag{15}$$

Next, in the following Lemma 6, we establish an upper bound on the right-hand side of the inequality in Lemma 5, using Assumption 3 and the previously obtained Lemma 3.

**Lemma 6** ($\downarrow$). *Under the conditions of Theorem 1, the following inequality holds:*

$$\begin{aligned}
\langle \mathbf{I}, \mathrm{proj}_{\mathcal{H}}(\mathbb{E}[\mathbf{S}_K])^{1/2}\rangle \leq{}& \sqrt{K+1}^{\frac{1-\nu}{1+\nu}} \|\mathbf{L}\|_{\mathrm{tr}}^{\frac{1}{1+\nu}} \left[\sum_{k=0}^{K} \mathbb{E}[f(x_k) - f(x^*)]\right]^{\frac{\nu}{1+\nu}} \\
&+ \sqrt{K+1}\|\mathbf{\Sigma}\|_{\mathrm{tr}}.
\end{aligned} \tag{16}$$

Finally, with the help of Lemmas 5 and 6, we obtain the convergence result for Algorithm 1 in the following Theorem 1. Note that this result requires the inequality in eq. (17) to hold almost surely, which may not be satisfied, especially in the stochastic setting. However, this issue can be easily resolved with an additional projection step at each iteration. Refer to Appendix D for details.

**Theorem 1** ($\downarrow$). *Under Assumptions 1, 2 and 3, let $\eta = \mathcal{R}$, where $\mathcal{R} > 0$ almost surely satisfies the following inequality:*

$$\max_{k=0,\dots,K} \mathcal{R}(x_k - x^*) \leq \mathcal{R}. \tag{17}$$

*Then, the output $\overline{x}_K \in \mathcal{X}$ of Algorithm 1 satisfies the following inequality:*

$$\mathbb{E}[f(\overline{x}_K) - f(x^*)] \leq \frac{3\|\mathbf{L}\|_{\mathrm{tr}}\mathcal{R}^{1+\nu}}{(K+1)^{\frac{1+\nu}{2}}} + \frac{3\|\mathbf{\Sigma}\|_{\mathrm{tr}}\mathcal{R}}{\sqrt{K+1}} + \frac{3\sqrt{\delta}\mathcal{R}\dim(\mathcal{X})}{(K+1)}. \tag{18}$$

## 3.2 RELATED ALGORITHMS

In this section, we discuss the connection of Algorithm 1 with existing adaptive gradient methods with preconditioning.

**Connection with AdaGrad-Norm, AdaGrad, and ASGO/One-sided Shampoo.** We can obtain AdaGrad-Norm, AdaGrad, ASGO/One-sided Shampoo as special instances of Algorithm 1 by choosing the space of preconditioning operators $\mathcal{H}$ satisfying Assumption 1 according to Table 1. In the case $\nu = 1$, Theorem 1 recovers the state-of-the-art convergence guarantees for AdaGrad under anisotropic smoothness (Liu et al., 2024b) and for ASGO/One-sided Shampoo (An et al., 2025; Xie et al., 2025) under matrix smoothness. However, recall that Liu et al. (2024b); An et al. (2025); Xie et al. (2025) require a more restrictive noise variance bound as discussed in Section 2.2, and do not cover Hölder smoothness. In contrast, Theorem 1 works for arbitrary $\nu \in [0, 1]$, which implies that Algorithm 1 can adapt to different levels of anisotropic/matrix smoothness.

**Connection with DASGO.** Notably, Algorithm 1 recovers DASGO, a lightweight version of ASGO/One-sided Shampoo that uses diagonal preconditioning and was proposed by An et al. (2025) without any convergence guarantees. Consequently, Theorem 1 provides the first convergence guarantees for DASGO, to the best of our knowledge. Moreover, in Section 4, we will show that the convergence of DASGO, as well as AdaGrad, can be accelerated using Nesterov momentum.

**Connection between ASGO/One-sided Shampoo and Muon.** Recently, Jordan et al. (2024) proposed using the Shampoo optimizer (Gupta et al., 2018) with gradient accumulation turned off. This led to the development of Muon, a new optimizer with promising practical performance. The convergence of Muon was analyzed from the perspective of gradient methods with the non-Euclidean matrix spectral norm by Bernstein & Newhouse (2024); Pethick et al. (2025); Kovalev (2025). Notably, our analysis captures the connection between ASGO/One-sided Shampoo and non-Euclidean optimization with the spectral norm. Indeed, as discussed in Section 2.2, Assumption 2 implies the $(\|\mathbf{L}\|_{\mathrm{tr}}, \nu)$-Hölder smoothness in eq. (13) with respect to the norm $\mathcal{R}(\cdot)$, which, according to Table 1, coincides with the matrix spectral norm (up to constant factors). Moreover, in the case of ASGO/One-sided Shampoo, Theorem 1 provides the convergence result in terms of the constant $\|\mathbf{L}\|_{\mathrm{tr}}$ and the norm $\mathcal{R}(\cdot)$.

**Connection between DASGO and Scion.** Recently, Pethick et al. (2025) proposed Scion, a new variant of Muon, which, instead of the spectral norm, can use the matrix norm $\|\cdot\|_{2\to\infty}$: the maximal Euclidean norm of a row of a matrix. Note that in the case of DASGO, the norm $\mathcal{R}(\cdot)$ defined in eq. (12) coincides with the norm $\|\cdot\|_{2\to\infty}$ up to multiplicative constants, according to Table 1. Hence, Scion with the norm $\|\cdot\|_{2\to\infty}$ can be obtained by turning off the gradient accumulation in DASGO, that is, choosing $\mathbf{S}_k = g_k\langle g_k, \cdot\rangle$ in eq. (5). In other words, DASGO is connected to Scion in the same way as ASGO/(One-sided) Shampoo is connected to Muon. It is important to highlight that the iterations of Shampoo are not cheap and require matrix inversions, which triggered the development of the computationally effective alternative, Muon, by Jordan et al. (2024). However, the iterations of DASGO are not only inexpensive, but they also utilize adaptive preconditioning and have much more attractive theoretical convergence properties compared to Scion. Hence, it is worth trying to use DASGO in the practical scenarios identified by Pethick et al. (2025) to benefit from using the non-Euclidean norm $\|\cdot\|_{2\to\infty}$.

## 4 SGD WITH PRECONDITIONING AND ACCELERATION

### 4.1 GENERAL ACCELERATED ALGORITHM AND ITS CONVERGENCE

In this section, we develop accelerated adaptive SGD with preconditioning, which is summarized in Algorithm 2, and provide its unified convergence analysis. First, to simplify the analysis, we use the interpretation of Nesterov momentum acceleration (Nesterov, 1983) by Kovalev & Borodich (2024). The idea is that we define the functions $f_k(x)\colon \mathcal{X} \to \mathbb{R}$ as follows:

$$f_k(x) = \alpha_k^{-2} \cdot f(\alpha_k x + (1 - \alpha_k)\overline{x}_k), \quad \text{where } \alpha_k \in (0, 1] \text{ and } \overline{x}_k \in \mathcal{X}, \tag{19}$$

where $\overline{x}_k \in \mathcal{X}$ is updated according to line 7 at each iteration. We then apply the preconditioned SGD iterations in eq. (3) to this "time-varying" function $f_k(x)$. With this approach, we can upper-

---

**Algorithm 2** Accelerated Adaptive SGD with Preconditioning

---

1: **input:** $x_0 = \overline{x}_0 \in \mathcal{X}$, $K \in \{1, 2, \dots\}$
2: **for** $k = 0, \dots, K$ **do**
3:     sample $\xi_k \sim \mathcal{D}$
4:     compute $g_k = \nabla f_k(x_k; \xi_k)$, where $f_k(x)$ is defined in eq. (19)
5:     compute $\mathbf{H}_k \in \mathcal{H} \cap \mathbb{S}_{++}$ using eqs. (5) and (8)
6:     compute $x_{k+1} \in \mathcal{X}$ using eq. (3).
7:     compute $\overline{x}_{k+1} = \alpha_k x_{k+1} + (1 - \alpha_k)\overline{x}_k$
8: **output:** $\overline{x}_{K+1}$

---

bound the expected objective function suboptimality $\mathbb{E}[f(\overline{x}_{K+1}) - f(x^*)]$ using the expected regret-like sum $\sum_{k=0}^{K} \mathbb{E}[f_k(x_{k+1}) - f_k(x^*)]$ in the following Lemma 7.

**Lemma 7** ($\downarrow$). *Under the conditions of Theorem 2, the following inequality holds:*

$$\tfrac{1}{4}(K+2)^2 \mathbb{E}[f(\overline{x}_{K+1}) - f(x^*)] \leq \sum_{k=0}^{K} \mathbb{E}[f_k(x_{k+1}) - f_k(x^*)]. \tag{20}$$

Next, we proceed with the additional Assumption 4 on the operators $\mathbf{L}, \mathbf{\Sigma} \in \mathcal{H}$ defined in Assumptions 2 and 3. It is important to highlight that this assumption always holds when the space of preconditioners $\mathcal{H}$ contains only diagonal operators. Hence, this assumption is automatically satisfied for algorithms with diagonal preconditioning like AdaGrad and DASGO.

**Assumption 4.** *The operators $\mathbf{L} \in \mathcal{H}$ in Assumption 2 and $\mathbf{\Sigma} \in \mathcal{H}$ in Assumption 3 commute with the space $\mathcal{H}$, that is, $\mathbf{L}\mathbf{H} = \mathbf{H}\mathbf{L}$ and $\mathbf{\Sigma}\mathbf{H} = \mathbf{H}\mathbf{\Sigma}$ for all $\mathbf{H} \in \mathcal{H}$.*

The key idea for the analysis of Algorithm 2 is that under Assumption 4, the square of the precondition operator $\mathbf{H}_k$, defined in eq. (8), is a solution to the optimization problem in eq. (21), as indicated by Lemma 8. Hence, similar to the analysis of the non-accelerated Algorithm 1, we can utilize the FTL-BTL lemma (FTL-BTL) and obtain one of the key inequalities in Lemma 9.

**Lemma 8** ($\downarrow$). *Under Assumption 4, the operator $\mathbf{H}_k^2$ defined by eq. (8) is a solution to the following problem, where $\mathbf{B} = \mathbf{L}$ or $\mathbf{B} = \mathbf{\Sigma}$:*

$$\mathbf{H}_k^2 \in \underset{\mathbf{Q} \in \mathcal{H} \cap \mathbb{S}_{++}}{\arg\min} \langle \mathbf{Q}, \mathbf{B}\mathbf{S}_k \rangle + \langle \mathbf{B}, \delta\mathbf{Q} - \eta^2 \ln(\mathbf{Q}) \rangle. \tag{21}$$

**Lemma 9** ($\downarrow$). *Under Assumption 4, the following inequality holds for $\mathbf{B} = \mathbf{L}$ or $\mathbf{B} = \mathbf{\Sigma}$:*

$$\mathbb{E}\left[\sum_{i=0}^{k} \|g_i\|_{\mathbf{B}\mathbf{H}_i^2}^2\right] \leq \eta^2 \|\mathbf{B}\|_{\mathrm{tr}} \ln\left[\tfrac{1}{\delta}\eta^2 \left(\mathbb{E}[\|\mathbf{H}_k^{-1}\|_{\mathrm{tr}}]\right)^2\right]. \tag{22}$$

Finally, using the inequality in Lemma 9, we obtain the key upper bound on the regret-like sum $\mathbb{E}[f_k(x_{k+1}) - f_k(x^*)]$ in Lemma 10.

**Lemma 10** ($\downarrow$). *Under the conditions of Theorem 2, the following inequality holds:*

$$\sum_{k=0}^{K} \mathbb{E}[f_k(x_{k+1}) - f_k(x^*)] \leq \tfrac{1}{4}\mathcal{C}_K (K+2)^{\frac{3(1-\nu)}{2}} \|\mathbf{L}\|_{\mathrm{tr}} \mathcal{R}^{1+\nu}$$
$$+ \tfrac{1}{4}\mathcal{C}_K (K+2)^{\frac{3}{2}} \|\mathbf{\Sigma}\|_{\mathrm{tr}} \mathcal{R} + \sqrt{\delta}\mathcal{R}\dim(\mathcal{X}). \tag{23}$$

Now, all that remains is to combine Lemma 10 with Lemma 7 and obtain the main convergence result for Algorithm 2 in Theorem 2. Similar to the non-accelerated result in Theorem 1, we require the inequality in eq. (17) to hold almost surely. This can be easily guaranteed by an additional projection step at each iteration, as discussed in Appendix D.

**Theorem 2.** *Under Assumptions 1, 2, 3 and 4, let $\eta = 2\mathcal{R}$, where $\mathcal{R} > 0$ satisfies eq. (17), and let $\alpha_k = 2/(k+2)$. Then, the output $\overline{x}_{K+1} \in \mathcal{X}$ of Algorithm 2 satisfies the following inequality:*

$$\mathbb{E}[f(\overline{x}_{K+1}) - f(x^*)] \leq \frac{\mathcal{C}_K \|\mathbf{L}\|_{\mathrm{tr}} \mathcal{R}^{1+\nu}}{(K+2)^{\frac{1+3\nu}{2}}} + \frac{\mathcal{C}_K \|\mathbf{\Sigma}\|_{\mathrm{tr}} \mathcal{R}}{\sqrt{K+2}} + \frac{4\sqrt{\delta}\mathcal{R}\dim(\mathcal{X})}{(K+2)^2}, \tag{24}$$

*where the constant $\mathcal{C}_K > 0$ satisfies the following relation:*

$$\mathcal{C}_K = \mathcal{O}\left(1 + \ln K + \ln \frac{\|\mathbf{L}\|_{\mathrm{tr}} \mathcal{R}^\nu}{\sqrt{\delta}} + \ln \frac{\|\mathbf{\Sigma}\|_{\mathrm{tr}}}{\sqrt{\delta}}\right). \tag{25}$$

### 4.2 ADAGRAD AND DASGO WITH MOMENTUM ACCELERATION

In this section, we provide a detailed discussion of our results for two special instances of adaptive gradient methods with diagonal preconditioning: AdaGrad and DASGO. In the case of DASGO, let $\mathcal{X} = \mathbb{R}^{m \times n}$ be the space of $m \times n$ matrices and consider the following special instance of problem (1):

$$\min_{X \in \mathbb{R}^{m \times n}} f(X). \tag{26}$$

We choose the space $\mathcal{H}$ of preconditioning operators $\mathbb{R}^{m \times n} \to \mathbb{R}^{m \times n}$ for DASGO according to Table 1. That is, $\mathcal{H} = \{G \mapsto \mathrm{diag}(\boldsymbol{b})G : \boldsymbol{b} \in \mathbb{R}^m\}$, which obviously satisfies Assumption 1. Note that AdaGrad can be obtained from DASGO by simply choosing $n = 1$. Henceforth, for simplicity, we will consider only DASGO.

Next, we specialize Assumptions 2 and 3 to the setting of DASGO. In particular, we define the operator $\mathbf{L} \in \mathcal{H}$ in Assumption 2 as $\mathbf{L}: X \mapsto n^{\frac{\nu-1}{2}} \mathrm{diag}(\boldsymbol{l})X$, where $\boldsymbol{l} = (\boldsymbol{l}_1, \ldots, \boldsymbol{l}_m) \in \mathbb{R}_{++}^m$ and $X \in \mathbb{R}^{m \times n}$. For example, in the case $\nu = 1$ and $n = 1$, Assumption 2 exactly matches the anisotropic smoothness assumption used by Liu et al. (2024b). In the general case $\nu \in [0, 1]$ and $n \geq 1$, Assumption 2 implies the $(\|\boldsymbol{l}\|_1, \nu)$-Hölder smoothness with respect to the non-Euclidean norm $\|\cdot\|_{2\to\infty}$, that is, the following special instance of the inequality in eq. (13) holds:

$$0 \leq f(X_2) - f(X_1) - \langle \nabla f(X_1), X_2 - X_1 \rangle \leq \frac{1}{1+\nu} \|\boldsymbol{l}\|_1^{\frac{1-\nu}{2}} \|X_2 - X_1\|_{2\to\infty}^{1+\nu}. \tag{27}$$

Similarly, we define the operator $\boldsymbol{\Sigma} \in \mathcal{H}$ in Property A3.2 as $\boldsymbol{\Sigma}: X \mapsto n^{-\frac{1}{2}} \mathrm{diag}(\boldsymbol{\sigma})X$, where $\boldsymbol{\sigma} = (\boldsymbol{\sigma}_1, \ldots, \boldsymbol{\sigma}_m) \in \mathbb{R}_{++}^m$ and $X \in \mathbb{R}^{m \times n}$. Consequently, the variance bound in Property A3.2 turns into the following inequality:

$$\mathbb{E}_{\xi \sim \mathcal{D}}\left[\sum_{i=1}^m (1/\boldsymbol{\sigma}_i)\|N_i\|^2\right] \leq \|\boldsymbol{\sigma}\|_1, \quad \text{where} \quad [N_1, \ldots, N_m]^\top = \nabla F(X; \xi) - \nabla F(X). \tag{28}$$

This inequality is implied, for instance, by the anisotropic noise assumption used by Liu et al. (2024b), and hence, it is more general.

Further, for simplicity in the presentation of the results, we use the convergence guarantees from Appendix D for the algorithms with projection steps. Using Theorem 3 and assuming $\delta \ll 1$, we obtain the following convergence guarantees for AdaGrad and DASGO:

$$\mathbb{E}[f(\overline{X}_K) - f(X^*)] \leq \tilde{\mathcal{O}}\left(\frac{\|\boldsymbol{l}\|_1\|X^*\|_{2\to\infty}^{1+\nu}}{K^{\frac{1+\nu}{2}}} + \frac{\|\boldsymbol{\sigma}\|_1\|X^*\|_{2\to\infty}}{\sqrt{K+1}}\right). \tag{29}$$

This matches the result of Liu et al. (2024b) for AdaGrad in the smooth case ($\nu = 1$ and $n = 1$), but also provides convergence guarantees for DASGO. Similarly, using Theorem 4, we establish convergence guarantees for AdaGrad and DASGO with Nesterov momentum:

$$\mathbb{E}[f(\overline{X}_{K+1}) - f(X^*)] \leq \tilde{\mathcal{O}}\left(\frac{\|\boldsymbol{l}\|_1\|X^*\|_{2\to\infty}^{1+\nu}}{K^{\frac{1+3\nu}{2}}} + \frac{\|\boldsymbol{\sigma}\|_1\|X^*\|_{2\to\infty}}{\sqrt{K+1}}\right), \tag{30}$$

which substantially improves upon the non-accelerated result above. We can also compare this result with the state-of-the-art result of Kavis et al. (2019); Rodomanov et al. (2024) for scalar AdaGrad-type stepsizes under the above assumptions:

$$\mathbb{E}[f(\overline{X}_{K+1}) - f(X^*)] \leq \tilde{\mathcal{O}}\left(\frac{\|\boldsymbol{l}\|_\infty\|X^*\|^{1+\nu}}{K^{\frac{1+3\nu}{2}}} + \frac{\sqrt{m}\|\boldsymbol{\sigma}\|_\infty\|X^*\|}{\sqrt{K+1}}\right). \tag{31}$$

Our result in eq. (30) is substantially better than the existing result in eq. (31) as long as $\|\boldsymbol{l}\|_1 \sim \|\boldsymbol{l}\|_\infty$, $\|\boldsymbol{\sigma}\|_1 \sim \|\boldsymbol{\sigma}\|_\infty$, and $\|X^*\| \gg \|X^*\|_{2\to\infty}$. For instance, in the AdaGrad case ($n = 1$), this holds when $\boldsymbol{l}$ and $\boldsymbol{\sigma}$ are sparse and $X^*$ is dense, which aligns with the conclusions made by Liu et al. (2024b) for AdaGrad without momentum acceleration.

### ACKNOWLEDGEMENTS

This work was supported by the The Ministry of Economic Development of the Russian Federation in accordance with the subsidy agreement (agreement identifier 000000C313925P4G0002; grant No 139-15-2025-011).

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

# Appendix

## A  NOTATION

In this paper, we use the following notation: $\dim(\mathcal{X})$ is the dimension of the space $\mathcal{X}$; $\mathbb{L}$ is the space of linear operators $\mathcal{X} \to \mathcal{X}$, for arbitrary operator $\mathbf{A} \in \mathbb{L}$, $\mathbf{A}^* \in \mathbb{L}$ denotes its adjoint operator, $\mathbf{I} \in \mathbb{L}$ and $\mathbf{O} \in \mathbb{L}$ denote the identity and the zero operators, respectively; $\mathbb{S} \subset \mathbb{L}$ is the space of self-adjoint linear operators, $\mathbb{S}_{++}, \mathbb{S}_+ \subset \mathbb{S}$ are the spaces of positive definite and positive semi-definite self-adjoint operators, respectively; $\prec, \preceq, \succ, \succeq$ denote the standard Löwner order on $\mathbb{S}$; $\langle \cdot, \cdot \rangle$ and $\| \cdot \|$ denote the standard inner product and Euclidean norm on $\mathcal{X}$ or $\mathbb{L}$, depending on the context, in particular, $\langle \mathbf{A}, \mathbf{B} \rangle = \mathrm{tr}(\mathbf{A}\mathbf{B}^*)$ for $\mathbf{A}, \mathbf{B} \in \mathbb{L}$; for arbitrary $\mathbf{H} \in \mathbb{S}_{++}$, $\| \cdot \|_{\mathbf{H}}$ denotes the weighted Euclidean norm in $\mathcal{X}$, i.e., $\|x\|_{\mathbf{H}}^2 = \langle x, \mathbf{H}x \rangle$ for $x \in \mathcal{X}$; $\| \cdot \|_{\mathrm{op}}$ and $\| \cdot \|_{\mathrm{tr}}$ denote the operator and trace norm on $\mathbb{L}$, respectively, i.e., $\|\mathbf{A}\|_{\mathrm{op}} = \max_{\|x\| \le 1} \|\mathbf{A}x\|$ and $\|\mathbf{A}\|_{\mathrm{tr}} = \mathrm{tr}(\sqrt{\mathbf{A}\mathbf{A}^*})$ for all $\mathbf{A} \in \mathbb{L}$; for arbitrary $y, z \in \mathcal{X}$, by $z\langle y, \cdot \rangle \in \mathbb{L}$ we denote the rank-1 linear operator $x \mapsto \langle x, y \rangle z$; by $\mathbb{S}^d \subset \mathbb{R}^{d \times d}$ we denote the space of $d \times d$ symmetric matrices; by $\odot$, we denote the Hadamard vector or matrix product.

## B  ADDITIONAL RELATED WORK

**Exponential moving average.** AdaGrad-type algorithms, like RMSProp, often utilize the exponential moving average (EMA): they replace the cumulative sum of the squared gradients $\sum_{i=0}^k \|g_i\|^2$ in eq. (2) with the exponential moving average $\sum_{i=0}^k \beta^i \|g_i\|^2$. Notably, EMA is the third key component of Adam, in addition to diagonal preconditioning and momentum. Moreover, Défossez et al. (2020) showed how to analyze AdaGrad with EMA and explained that it is related to the standard AdaGrad in the same way as fixed stepsize SGD is related to decaying stepsize SGD. Consequently, we can develop EMA versions of our algorithms as well as their convergence proofs. However, Défossez et al. (2020) could not justify the benefits of using momentum. Hence, our theoretical justification of the benefits of momentum and diagonal preconditioning, combined with the analysis of EMA by Défossez et al. (2020), may provide the ultimate explanation for the efficiency of Adam.

**Parameter-free algorithms.** There is an important research direction aimed at designing parameter-free variants of AdaGrad, which can avoid tuning the parameter $\eta \propto \|x^*\|$ in eq. (2). This includes the works of Cutkosky & Orabona (2018); Orabona & Pál (2021); Defazio & Mishchenko (2023); Mishchenko & Defazio (2023); Ivgi et al. (2023); Khaled et al. (2023); Kreisler et al. (2024).[4] However, to the best of our knowledge, the existing results are applicable only to scalar stepsizes, which are rarely used in practice. Designing parameter-free gradient methods with diagonal or matrix preconditioning is an interesting question for future work.

**Concurrent unified analysis framework.** Xie et al. (2025) developed a unified analysis for AdaGrad-type methods, where they also adopt the matrix smoothness assumption. We found their work during the preparation of our literature review, at a point when our results had already been finalized. Although the results of Xie et al. (2025) share some similarities with ours and are capable of providing a partially positive answer to Question 1, their analysis has substantial differences and drawbacks. Specifically, it only covers the smooth case and lacks adaptation to non-smooth or Hölder smooth functions. In addition, it requires a more restrictive stochastic gradient noise assumption and, most importantly, does not contain any results about using momentum acceleration, thus completely missing an answer to the fundamental Question 2.

## C  MOTIVATION FOR CONVEX SETTING

In this paper, we focus on the case where the objective function $f(x)$ in problem (1) is convex. There are multiple reasons for this assumption. First, optimization algorithms for convex functions hold substantial practical interest because empirical studies (Zhou et al., 2019; Kleinberg et al., 2018) suggest that deep neural networks may adhere to convexity or its variants. Second, for gen-

---

[4]Additional references can be found in the overview of Orabona (2023).

---

**Algorithm 3** Adaptive SGD with Preconditioning and Weight Clipping

1: **input:** $x_0 \in \mathcal{Q}_{\mathcal{R}}$, $K \in \{1, 2, \ldots\}$
2: **for** $k = 0, \ldots, K$ **do**
3: $\quad$ sample $\xi_k \sim \mathcal{D}$
4: $\quad$ compute $g_k = \nabla f(x_k; \xi_k)$
5: $\quad$ compute $\mathbf{H}_k \in \mathcal{H} \cap \mathbb{S}_{++}$ using eqs. (5) and (8)
6: $\quad$ compute $x_{k+1} \in \mathcal{X}$ using eq. (32)
7: **output:** $\overline{x}_K = \frac{1}{K+1} \sum_{k=0}^{K} x_k$

---

**Algorithm 4** Accelerated Adaptive SGD with Preconditioning and Weight Clipping

1: **input:** $x_0 = \overline{x}_0 \in \mathcal{Q}_{\mathcal{R}}$, $K \in \{1, 2, \ldots\}$
2: **for** $k = 0, \ldots, K$ **do**
3: $\quad$ sample $\xi_k \sim \mathcal{D}$
4: $\quad$ compute $g_k = \nabla f_k(x_k; \xi_k)$, where $f_k(x)$ is defined in eq. (19)
5: $\quad$ compute $\mathbf{H}_k \in \mathcal{H} \cap \mathbb{S}_{++}$ using eqs. (5) and (8)
6: $\quad$ compute $x_{k+1} \in \mathcal{X}$ and $x_{k+1/2} \in \mathcal{X}$ using eq. (32)
7: $\quad$ compute $\overline{x}_{k+1} = \alpha_k x_{k+1/2} + (1 - \alpha_k)\overline{x}_k$
8: **output:** $\overline{x}_{K+1}$

---

eral non-convex functions, it is impossible to achieve meaningful global convergence beyond vague first-order stationarity (Carmon et al., 2020). However, in practice, it is typically desirable to achieve small values of the objective function, which can only be guaranteed under additional assumptions, such as gradient domination (Fatkhullin et al., 2022), star/quasar convexity (Hinder et al., 2020), etc. Such assumptions are, in turn, relaxations of the convexity property itself. Hence, it is natural to consider the convex setting first before trying to relax it. Finally, convex optimization serves as a large source of inspiration for designing efficient optimization algorithms. Notably, many optimization techniques that have practical benefits were initially theoretically justified for convex functions. These include momentum acceleration (Nesterov, 2013), local training (Mishchenko et al., 2022), and AdaGrad (Duchi et al., 2011), on which Adam itself is based.

## D ALGORITHMS WITH WEIGHT CLIPPING

The upper bounds on the expected functional suboptimality in Theorem 1 for Algorithm 1 and in Theorem 2 for Algorithm 2 require the inequality in eq. (17) to hold almost surely. However, this requirement may not be satisfied, for instance, in the stochastic case. It is important to higlight that such issue is not an artifact of our analysis but a common phenomenon in AdaGrad-type algorithms (Duchi et al., 2011; Gupta et al., 2018; Liu et al., 2024b; An et al., 2025; Xie et al., 2025). To bypass this issue, a typical approach is to modify the preconditioned gradient update rule in eq. (3) by adding an extra projection step onto the set $\mathcal{Q}_{\mathcal{R}} = \{x \in \mathcal{X} : \mathcal{R}(x) \leq \mathcal{R}\}$, where $\mathcal{R} > \mathcal{R}(x^*)$. The modified update rule is given as follows:

$$x_{k+1} = \arg\min_{x \in \mathcal{Q}_{\mathcal{R}}} \tfrac{1}{2}\|x - x_{k+1/2}\|^2_{\mathbf{H}_k^{-1}}, \quad x_{k+1/2} = \arg\min_{x \in \mathcal{X}} \langle g_k, x\rangle + \tfrac{1}{2}\|x - x_k\|^2_{\mathbf{H}_k^{-1}}. \quad (32)$$

Note that the set $\mathcal{Q}_{\mathcal{R}}$ is convex and hence the projection step is well-defined. Also, note that the projection is performed with respect to the weighted Euclidean norm $\|\cdot\|_{\mathbf{H}_k^{-1}}$, which may be expensive, for instance, when the preconditioner $\mathbf{H}_k$ is dense. However, this projection step can be computed efficiently when the preconditioner $\mathbf{H}_k$ is diagonal. For instance, the projection is equivalent to the coordinate-wise clipping $t \mapsto \min\{\mathcal{R}, \max\{-\mathcal{R}, t\}\}$ in AdaGrad, and to the row-wise or column-wise norm-clipping $z \mapsto \min\{1, \mathcal{R}/\|z\|\}z$ in DASGO. Below, we discuss the modified update rule eq. (32) in relation to the non-accelerated Algorithm 1 and the accelerated Algorithm 2 in detail, including the additional modifications in Algorithms 1 and 2 and the modifications in the convergence proofs.

**Non-accelerated Algorithm 1 $\rightarrow$ Algorithm 3.** The only modifications to Algorithm 1 are the initialization $x_0 \in \mathcal{Q}$ on line 1 and the modified update rule (32) on line 6 in Algorithm 3, as

discussed above. We also modify the proof of Lemma 5 in Appendix F.1 by obtaining the following:

$$\frac{1}{2}\|x_{k+1} - x^*\|^2_{\mathbf{H}_k^{-1}} \overset{(a)}{\leq} \frac{1}{2}\|x_{k+1/2} - x^*\|^2_{\mathbf{H}_k^{-1}} \overset{(b)}{=} \frac{1}{2}\|x_k - x^*\|^2_{\mathbf{H}_k^{-1}} - \langle g_k, x_k - x^* \rangle + \frac{1}{2}\|g_k\|^2_{\mathbf{H}_k}, \quad (33)$$

where (a) uses the update rule for $x_{k+1}$ in eq. (32), the non-expansiveness of the projection, and the fact that $x^* \in \mathcal{Q}_\mathcal{R}$; (b) uses the update rule for $x_{k+1/2}$ in eq. (32). One can observe that this eq. (33) coincides with eq. (43) in Appendix F.1. Moreover, the inequality eq. (17) holds almost surely due to the projection step in eq. (32). Therefore, the rest of the proof of Theorem 1 remains unchanged, and we obtain the following Theorem 3.

**Theorem 3.** *Under Assumptions 1, 2 and 3, let $\eta = \mathcal{R}$, where $\mathcal{R} > \mathcal{R}(x^*)$. Then, the output $\overline{x}_K \in \mathcal{X}$ of Algorithm 3 satisfies the following inequality:*

$$\mathbb{E}[f(\overline{x}_K) - f(x^*)] \leq \frac{3\|\mathbf{L}\|_{\mathrm{tr}}\mathcal{R}^{1+\nu}}{(K+1)^{\frac{1+\nu}{2}}} + \frac{3\|\mathbf{\Sigma}\|_{\mathrm{tr}}\mathcal{R}}{\sqrt{K+1}} + \frac{3\sqrt{\delta}\mathcal{R}\dim(\mathcal{X})}{(K+1)}. \quad (34)$$

**Accelerated Algorithm 2 $\to$ Algorithm 4.** Similarly, to the non-accelerated algorithm, the accelerated Algorithm 4 contains the modified initialization $x_0 = \overline{x}_0 \in \mathcal{Q}_\mathcal{R}$ on line 1 and the modified update rule (32) on line 6 in Algorithm 4. In addition to the modified eq. (33), we also modify the first inequality in the proof of Lemma 10 in Appendix G.4 as follows:

$$\mathbb{E}[f_k(x_{k+1/2})] \leq \mathbb{E}\left[ f_k(x_k) - \|g_k\|^2_{\mathbf{H}_k} + \langle n_k, \mathbf{H}_k g_k \rangle + \frac{1}{1+\nu}\alpha_k^{\nu-1}\|\mathbf{L}\|_{\mathrm{tr}}^{\frac{1-\nu}{2}}\|g_k\|^{1+\nu}_{\mathbf{L}\mathbf{H}_k^2} \right]. \quad (35)$$

Here, the only difference is the left-hand side $\mathbb{E}[f_k(x_{k+1/2})]$ compared to $\mathbb{E}[f_k(x_{k+1})]$ in Appendix G.4, which means that we also have to modify the update rule for $\overline{x}_{k+1}$ on line 7 of Algorithm 4 and apply trivial changes to Lemma 7. The rest of the proof of Theorem 2 remains unchanged and we obtain the following Theorem 4.

**Theorem 4.** *Under Assumptions 1, 2, 3 and 4, let $\eta = 2\mathcal{R}$, where $\mathcal{R} > \mathcal{R}(x^*)$, and let $\alpha_k = 2/(k+2)$. Then, the output $\overline{x}_{K+1} \in \mathcal{X}$ of Algorithm 4 satisfies the following inequality:*

$$\mathbb{E}[f(\overline{x}_{K+1}) - f(x^*)] \leq \frac{\mathcal{C}_K\|\mathbf{L}\|_{\mathrm{tr}}\mathcal{R}^{1+\nu}}{(K+2)^{\frac{1+3\nu}{2}}} + \frac{\mathcal{C}_K\|\mathbf{\Sigma}\|_{\mathrm{tr}}\mathcal{R}}{\sqrt{K+2}} + \frac{4\sqrt{\delta}\mathcal{R}\dim(\mathcal{X})}{(K+2)^2}, \quad (36)$$

*where the constant $\mathcal{C}_K > 0$ satisfies the following relation:*

$$\mathcal{C}_K = \mathcal{O}\left( 1 + \ln K + \ln \frac{\|\mathbf{L}\|_{\mathrm{tr}}\mathcal{R}^\nu}{\sqrt{\delta}} + \ln \frac{\|\mathbf{\Sigma}\|_{\mathrm{tr}}}{\sqrt{\delta}} \right). \quad (37)$$

# E    PROOFS FOR SECTION 2

## E.1    PROOF OF LEMMA 1

Let functions $l_{-1}(\mathbf{H}), \ldots, l_k(\mathbf{H}) \colon \mathbb{S}_{++} \to \mathbb{R}$ be defined as follows:

$$l_{-1}(\mathbf{H}) = \langle \mathbf{I}, \phi(\mathbf{H}) \rangle, \quad l_i(\mathbf{H}) = \|g_i\|_{\mathbf{H}}^2 \text{ for } i = 0, \ldots, k. \tag{38}$$

Let $\mathbf{H}_{-1} \in \mathcal{H} \cap \mathbb{S}_{++}$ be defined as follows:

$$\mathbf{H}_{-1} = \underset{\mathbf{H} \in \mathcal{H} \cap \mathbb{S}_{++}}{\arg\min} \langle \mathbf{I}, \phi(\mathbf{H}) \rangle. \tag{39}$$

From eq. (5), it is easy to verify that the following relation holds for all $i = -1, \ldots, k$:

$$\mathbf{H}_i = \underset{\mathbf{H} \in \mathcal{H} \cap \mathbb{S}_{++}}{\arg\min} \sum_{i=-1}^{k} l_i(\mathbf{H}).$$

Next, we get the following inequality:

$$\sum_{i=0}^{k} l_i(\mathbf{H}_i) \overset{(a)}{\leq} \sum_{i=-1}^{k} l_i(\mathbf{H}_i) \overset{(b)}{\leq} \sum_{i=-1}^{k} l_i(\mathbf{H}_k).$$

where (a) uses the assumption that the potential function $\phi(h)$ is non-negative; (b) uses eq. (FTL-BTL). It remains to to do rearranging and use the definition of the functions $l_i(\mathbf{H})$.    $\square$

## E.2    PROOF OF LEMMA 2

First, using Properties A1.1 and A1.2, we can show that $\mathbf{H}_k \in \mathcal{H} \cap \mathbb{S}_{++}$. Next, we show that $\mathbf{H}_k$ in eq. (8) is a solution to the problem in eq. (5) by verifying the first-order optimality condition:

$$\begin{aligned}
\nabla(\langle \cdot, \mathbf{S}_k \rangle + \langle \mathbf{I}, \phi(\cdot) \rangle)(\mathbf{H}_k) &\overset{(a)}{=} \mathbf{S}_k + \phi'(\mathbf{H}_k) \\
&\overset{(b)}{=} \mathbf{S}_k + \delta \mathbf{I} - \eta^2 \mathbf{H}_k^{-2} \\
&\overset{(c)}{=} \mathbf{S}_k - \text{proj}_{\mathcal{H}}(\mathbf{S}_k) \\
&\in \mathcal{H}^\perp.
\end{aligned}$$

where (a) uses the standard operator function calculus (Carlen, 2010); (b) uses eq. (7); (c) uses eq. (8). Next, we can show that the solution $\mathbf{H}_k$ is unique. Indeed, by Theorem 2.10 of Carlen (2010), the function $\langle \mathbf{I}, \phi(\mathbf{H}) \rangle$ is strictly convex, because the function $\phi(h)$ defined in eq. (7) is strictly convex. Finally, we can prove eq. (9). It follows from the operator monotonicity of the function $h \mapsto -1/\sqrt{h}$, which is implied by Löwner-Heinz Theorem (Carlen, 2010, Theorem 2.6), and the ordering $\text{proj}_{\mathcal{H}}(\mathbf{S}_{k+1}) \succeq \text{proj}_{\mathcal{H}}(\mathbf{S}_k)$, which is implied by Property A1.1 and the definition of $\mathbf{S}_k$ in eq. (5).    $\square$

## E.3    PROOF OF LEMMA 3

Assumption 2 implies the following inequality for all $x \in \mathcal{X}$ and $\nabla f(x) \in \partial f(x)$:

$$f(x^*) \leq f(x) + \langle \nabla f(x), x^* - x \rangle + \frac{1}{1+\nu} \|\mathbf{L}\|_{\text{tr}}^{\frac{1-\nu}{2}} \|x^* - x\|_{\mathbf{L}}^{1+\nu}. \tag{40}$$

In the case $\nu \in (0, 1]$, we can minimize the right-hand side in $x$, which gives the following:

$$\|\nabla f(x)\|_{\mathbf{L}^{-1}}^{\frac{1+\nu}{\nu}} \leq \left( \frac{1+\nu}{\nu} \right) \|\mathbf{L}\|_{\text{tr}}^{\frac{1-\nu}{2\nu}} \left( f(x) - f(x^*) \right). \tag{41}$$

Taking both sides in the power $\frac{2\nu}{1+\nu}$ gives the desired inequality in the case $\nu \in (0, 1]$. Finally, in the case $\nu = 0$, minimizing the right-hand side of the previous upper bound on $f(x^*)$ gives the following:

$$f(x^*) \leq f(x) + \begin{cases} 0 & \|\nabla f(x)\|_{\mathbf{L}^{-1}}^2 \leq \|\mathbf{L}\|_{\text{tr}} \\ -\infty & \|\nabla f(x)\|_{\mathbf{L}^{-1}}^2 > \|\mathbf{L}\|_{\text{tr}} \end{cases}. \tag{42}$$

It remains to use the fact that both $f(x)$ and $f(x^*)$ are finite to obtain the desired inequality in the case $\nu = 0$.    $\square$

### E.4 PROOF OF LEMMA 4

**(i) Non-negativity.** It is obvious.

**(ii) Absolute homogenity.** For arbitrary $t \in \mathbb{R}$ we can obtain the following:

$$\mathcal{R}(tx) \stackrel{(a)}{=} \|\text{proj}_{\mathcal{H}}(t^2\mathbf{X})\|_{\text{op}}^{1/2} \stackrel{(b)}{=} |t| \cdot \|\text{proj}_{\mathcal{H}}(\mathbf{X})\|_{\text{op}}^{1/2} \stackrel{(c)}{=} |t| \cdot \mathcal{R}(x),$$

where (a) and (c) use the definition of $\mathcal{R}(x)$ in eq. (12); (b) uses the linearity of the projection onto $\mathcal{H}$ and the absolute homogentiy of $\|\cdot\|_{\text{op}}$.

**(iii) Positive definiteness.** Let $\mathcal{R}(x) = 0$. Then $\text{proj}_{\mathcal{H}}(\mathbf{X}) = 0$, which implies the following:

$$0 = \langle \mathbf{I}, \text{proj}_{\mathcal{H}}(\mathbf{X}) \rangle \stackrel{(a)}{=} \langle \mathbf{I}, \mathbf{X} \rangle \stackrel{(b)}{=} \|x\|^2$$

where (a) uses the fact that $\mathbf{I} \in \mathcal{H}$ due to Property A1.2; (b) uses the definition of $\mathbf{X}$ in eq. (12). Hence, we get $x = 0$.

**(iv) Subadditivity.** Let $x, y \in \mathcal{X}$. Then we can obtain the following:

$$
\begin{aligned}
\mathcal{R}(x + y) &\stackrel{(a)}{=} \|\text{proj}_{\mathcal{H}}((x+y)\langle x+y, \cdot\rangle)\|_{\text{op}}^{1/2} \\
&\stackrel{(b)}{=} \|\text{proj}_{\mathcal{H}}((1+c^2)x\langle x, \cdot\rangle + (1+1/c^2)y\langle y, \cdot\rangle - (cx - y/c)\langle cx - y/c, \cdot\rangle)\|_{\text{op}}^{1/2} \\
&\stackrel{(c)}{\leq} \|(1+c^2)\text{proj}_{\mathcal{H}}(x\langle x, \cdot\rangle) + (1+1/c^2)\text{proj}_{\mathcal{H}}(y\langle y, \cdot\rangle)\|_{\text{op}}^{1/2} \\
&\stackrel{(d)}{\leq} ((1+c^2)\|\text{proj}_{\mathcal{H}}(x\langle x, \cdot\rangle)\|_{\text{op}} + (1+1/c^2)\|\text{proj}_{\mathcal{H}}(y\langle y, \cdot\rangle)\|_{\text{op}})^{1/2} \\
&\stackrel{(e)}{=} \|\text{proj}_{\mathcal{H}}(x\langle x, \cdot\rangle)\|_{\text{op}}^{1/2} + \|\text{proj}_{\mathcal{H}}(y\langle y, \cdot\rangle)\|_{\text{op}}^{1/2} \\
&\stackrel{(f)}{=} \mathcal{R}(x) + \mathcal{R}(y).
\end{aligned}
$$

where (a) and (f) use the definition of $\mathcal{R}(x)$ in eq. (12); (b) uses the bilinearity of the mapping $x \mapsto x\langle x, \cdot\rangle$ and an arbitrary constant $c \in \mathbb{R}$; (c) uses Property A1.1, the linearity of the projection onto $\mathcal{H}$, and the fact that $\|\cdot\|_{\text{op}}$ is order-preserving on $\mathbb{S}_+$; (d) uses the subadditivity and absolute homogenity of $\|\cdot\|_{\text{op}}$; (e) can be obtain by minimizing in $c$.

The proof is now complete. $\qquad\square$

# F PROOFS FOR SECTION 3

## F.1 PROOF OF LEMMA 5

Let $r_k = x_k - x^*$ and $\mathbf{R}_k = r_k \langle r_k, \cdot \rangle$. We can rewrite $\frac{1}{2}\|r_{k+1}\|^2_{\mathbf{H}_k^{-1}}$ as follows:

$$\frac{1}{2}\|r_{k+1}\|^2_{\mathbf{H}_k^{-1}} \overset{\text{(a)}}{=} \frac{1}{2}\|r_k\|^2_{\mathbf{H}_k^{-1}} - \langle g_k, r_k \rangle + \frac{1}{2}\|g_k\|^2_{\mathbf{H}_k}, \tag{43}$$

where (a) uses eq. (3). Next, we sum these equations for $k = 0, \ldots, K$ and get the following:

$$\sum_{k=0}^K \langle g_k, r_k \rangle$$
$$= \frac{1}{2}\sum_{k=0}^K \|g_k\|^2_{\mathbf{H}_k} + \frac{1}{2}\sum_{k=0}^K \left( \|r_k\|^2_{\mathbf{H}_k^{-1}} - \|r_{k+1}\|^2_{\mathbf{H}_k^{-1}} \right)$$
$$= \frac{1}{2}\sum_{k=0}^K \|g_k\|^2_{\mathbf{H}_k} + \frac{1}{2}\|r_0\|^2_{\mathbf{H}_0^{-1}} + \frac{1}{2}\sum_{k=1}^K \|r_k\|^2_{\mathbf{H}_k^{-1} - \mathbf{H}_{k-1}^{-1}} - \frac{1}{2}\|r_{K+1}\|^2_{\mathbf{H}_{K+1}^{-1}}$$
$$\leq \frac{1}{2}\sum_{k=0}^K \|g_k\|^2_{\mathbf{H}_k} + \frac{1}{2}\langle \mathbf{R}_0, \mathbf{H}_0^{-1} \rangle + \frac{1}{2}\sum_{k=1}^K \langle \mathbf{R}_k, \mathbf{H}_k^{-1} - \mathbf{H}_{k-1}^{-1} \rangle$$
$$\overset{\text{(a)}}{=} \frac{1}{2}\sum_{k=0}^K \|g_k\|^2_{\mathbf{H}_k} + \frac{1}{2}\langle \mathrm{proj}_{\mathcal{H}}(\mathbf{R}_0), \mathbf{H}_0^{-1} \rangle + \frac{1}{2}\sum_{k=1}^K \langle \mathrm{proj}_{\mathcal{H}}(\mathbf{R}_k), \mathbf{H}_k^{-1} - \mathbf{H}_{k-1}^{-1} \rangle$$
$$\overset{\text{(b)}}{\leq} \frac{1}{2}\sum_{k=0}^K \|g_k\|^2_{\mathbf{H}_k} + \frac{1}{2}\mathcal{R}^2\|\mathbf{H}_0^{-1}\|_{\mathrm{tr}} + \frac{1}{2}\mathcal{R}^2\sum_{k=1}^K \|\mathbf{H}_k^{-1} - \mathbf{H}_{k-1}^{-1}\|_{\mathrm{tr}}$$
$$\overset{\text{(c)}}{=} \frac{1}{2}\sum_{k=0}^K \|g_k\|^2_{\mathbf{H}_k} + \frac{1}{2}\mathcal{R}^2\langle \mathbf{I}, \mathbf{H}_0^{-1} \rangle + \frac{1}{2}\mathcal{R}^2\sum_{k=1}^K \langle \mathbf{I}, \mathbf{H}_k^{-1} - \mathbf{H}_{k-1}^{-1} \rangle$$
$$\overset{\text{(d)}}{\leq} \frac{1}{2}\langle \mathbf{H}_K, \mathbf{S}_K \rangle + \frac{1}{2}\langle \mathbf{I}, \phi(\mathbf{H}_K) \rangle + \frac{1}{2}\mathcal{R}^2\langle \mathbf{I}, \mathbf{H}_K^{-1} \rangle$$
$$\overset{\text{(e)}}{=} \frac{1}{2}\langle \mathbf{H}_K, \mathrm{proj}_{\mathcal{H}}(\mathbf{S}_K) \rangle + \frac{1}{2}\langle \mathbf{I}, \phi(\mathbf{H}_K) \rangle + \frac{1}{2}\mathcal{R}^2\langle \mathbf{I}, \mathbf{H}_K^{-1} \rangle$$

where (a) use the properties of the projection and the fact that $\mathbf{H}_k^{-1} \in \mathcal{H}$ due to Property A1.2 and eq. (8); (b) uses the Hölder's inequality for Schatten norms, the definition of the norm $\mathcal{R}(\cdot)$ in eq. (12), and the inequality in eq. (17); (c) uses the fact that $\mathbf{H}_{k+1}^{-1} \succeq \mathbf{H}_k^{-1}$, which is implied by eq. (9) and the operator monotonicity of the function $h \mapsto -1/h$, which is implied by Löwner-Heinz Theorem (Carlen, 2010, Theorem 2.6); (d) uses Lemma 1; (e) use the fact that $\mathbf{H}_k^{-1} \in \mathcal{H}$ due to Property A1.2 and eq. (8).

Next, using the definition of the potential function $\phi(\mathbf{H})$ in eq. (7), the expression for $\mathbf{H}_k$ in eq. (8), and the definition $\eta = \mathcal{R}$, we get the following inequality:

$$\sum_{k=0}^K \langle g_k, r_k \rangle \leq \frac{1}{2}\langle \mathbf{H}_K, \delta\mathbf{I} + \mathrm{proj}_{\mathcal{H}}(\mathbf{S}_K) \rangle + \frac{1}{2}(\eta^2 + \mathcal{R}^2)\langle \mathbf{I}, \mathbf{H}_K^{-1} \rangle$$
$$\overset{\text{(a)}}{=} \frac{3}{2}\mathcal{R}\langle \mathbf{I}, (\delta\mathbf{I} + \mathrm{proj}_{\mathcal{H}}(\mathbf{S}_K))^{1/2} \rangle,$$

where (a) uses the definition $\eta = \mathcal{R}$. After taking the expectation, recalling that $\xi_k$ is independent of $x_k$, and using Property A3.1, we get

$$\sum_{k=0}^K \mathbb{E}[\langle \nabla f(x_k), r_k \rangle] \leq \frac{3}{2}\mathcal{R}\mathbb{E}[\langle \mathbf{I}, (\delta\mathbf{I} + \mathrm{proj}_{\mathcal{H}}(\mathbf{S}_K))^{1/2} \rangle]$$
$$\overset{\text{(a)}}{\leq} \frac{3}{2}\mathcal{R}\langle \mathbf{I}, (\delta\mathbf{I} + \mathrm{proj}_{\mathcal{H}}(\mathbb{E}[\mathbf{S}_K]))^{1/2} \rangle$$
$$\overset{\text{(b)}}{\leq} \frac{3}{2}\mathcal{R}\langle \mathbf{I}, \mathrm{proj}_{\mathcal{H}}(\mathbb{E}[\mathbf{S}_K])^{1/2} \rangle + \frac{3}{2}\sqrt{\delta}\mathcal{R}\|\mathbf{I}\|_{\mathrm{tr}}$$
$$= \frac{3}{2}\mathcal{R}\langle \mathbf{I}, \mathrm{proj}_{\mathcal{H}}(\mathbb{E}[\mathbf{S}_K])^{1/2} \rangle + \frac{3}{2}\sqrt{\delta}\mathcal{R}\dim(\mathcal{X})$$

where (a) uses the concavity of the function $\mathbf{H} \mapsto \langle \mathbf{I}, \mathbf{H}^{1/2} \rangle$, which is implied by Theorem 2.10 of Carlen (2010), and the linearity of the projection onto $\mathcal{H}$; (b) uses the fact that function $\mathbf{H} \mapsto \langle \mathbf{I}, \mathbf{H}^{1/2} \rangle$ is subadditive for $\mathbf{H} \in \mathbb{S}_+$, which is implied by Lemma 3 of An et al. (2025). It remains to use the convexity property from Assumption 2. □

## F.2 PROOF OF LEMMA 6

Let $\mathbf{G}_k, \mathbf{N}_k \in \mathbb{S}_{++}$ be defined as follows:

$$\mathbf{G}_k = \sum_{i=0}^k \nabla f(x_k) \langle \nabla f(x_k), \cdot \rangle, \quad \mathbf{N}_k = \sum_{i=0}^k n(x_k; \xi_k) \langle n(x_k; \xi_k), \cdot \rangle. \tag{44}$$

Then, we can obtain the following:

$$\mathbb{E}[\mathbf{S}_k] \overset{(a)}{=} \sum_{i=0}^k \mathbb{E}[(\nabla f(x_k) + n(x_k; \xi_k))\langle \nabla f(x_k) + n(x_k; \xi_k), \cdot\rangle]$$

$$= \mathbb{E}[\mathbf{G}_k + \mathbf{N}_k] + \sum_{i=0}^k \mathbb{E}[\nabla f(x_k)\langle n(x_k; \xi_k), \cdot\rangle + n(x_k; \xi_k)\langle \nabla f(x_k), \cdot\rangle]$$

$$\overset{(b)}{=} \mathbb{E}[\mathbf{G}_k + \mathbf{N}_k],$$

where (a) uses the definition of $\mathbf{S}_k$ in eq. (5) and Assumption 3; (b) uses Property A3.1 and the fact that $\xi_k$ is independent of $x_k$. Using this, we obtain the following relation:

$$\langle \mathbf{I}, \mathrm{proj}_{\mathcal{H}}(\mathbb{E}[\mathbf{S}_k])^{1/2}\rangle = \langle \mathbf{I}, \mathrm{proj}_{\mathcal{H}}(\mathbb{E}[\mathbf{G}_k + \mathbf{N}_k])^{1/2}\rangle$$

$$\overset{(a)}{=} \langle \mathbf{I}, [\mathrm{proj}_{\mathcal{H}}(\mathbb{E}[\mathbf{G}_k]) + \mathrm{proj}_{\mathcal{H}}(\mathbb{E}[\mathbf{N}_k])]^{1/2}\rangle$$

$$\overset{(b)}{\leq} \langle \mathbf{I}, \mathrm{proj}_{\mathcal{H}}(\mathbb{E}[\mathbf{G}_k])^{1/2}\rangle + \langle \mathbf{I}, \mathrm{proj}_{\mathcal{H}}(\mathbb{E}[\mathbf{N}_k])^{1/2}\rangle,$$

where (a) uses the linearity of the expectation and the projection onto $\mathcal{H}$; (b) uses the fact that function $\mathbf{H} \mapsto \langle \mathbf{I}, \mathbf{H}^{1/2}\rangle$ is subadditive for $\mathbf{H} \in \mathbb{S}_+$, which is implied by Lemma 3 of An et al. (2025).

We can upper-bound $\langle \mathbf{I}, \mathrm{proj}_{\mathcal{H}}(\mathbb{E}[\mathbf{N}_k])^{1/2}\rangle$ as follows:

$$\langle \mathbf{I}, \mathrm{proj}_{\mathcal{H}}(\mathbb{E}[\mathbf{N}_k])^{1/2}\rangle = \langle \mathbf{\Sigma}^{1/2}, \mathbf{\Sigma}^{-1/2} \mathrm{proj}_{\mathcal{H}}(\mathbb{E}[\mathbf{N}_k])^{1/2}\rangle$$

$$\overset{(a)}{\leq} \|\mathbf{\Sigma}^{1/2}\|\|\mathbf{\Sigma}^{-1/2} \mathrm{proj}_{\mathcal{H}}(\mathbb{E}[\mathbf{N}_k])^{1/2}\|$$

$$\overset{(b)}{=} \sqrt{\|\mathbf{\Sigma}\|_{\mathrm{tr}}\langle \mathbf{\Sigma}^{-1}, \mathrm{proj}_{\mathcal{H}}(\mathbb{E}[\mathbf{N}_k])\rangle}$$

$$\overset{(c)}{=} \sqrt{\|\mathbf{\Sigma}\|_{\mathrm{tr}}\mathbb{E}[\langle \mathbf{\Sigma}^{-1}, \mathbf{N}_k\rangle]}$$

$$\overset{(d)}{\leq} \sqrt{\|\mathbf{\Sigma}\|_{\mathrm{tr}}\sum_{i=0}^k \mathbb{E}[\|n(x_i; \xi_i)\|_{\mathbf{\Sigma}^{-1}}^2]}$$

$$\overset{(e)}{\leq} \sqrt{k+1}\|\mathbf{\Sigma}\|_{\mathrm{tr}}$$

where (a) uses the Cauchy-Schwarz inequality; (b) uses the definition of $\|\cdot\|$ and $\|\cdot\|_{\mathrm{tr}}$; (c) uses the linearity of the expectation and the fact that $\mathbf{\Sigma}^{-1} \in \mathcal{H}$, which is implied by Properties A1.2 and A3.2; (d) uses the definition of $\mathbf{N}_k$; (e) uses Property A3.2.

Similarly, we can upper-bound $\langle \mathbf{I}, \mathrm{proj}_{\mathcal{H}}(\mathbb{E}[\mathbf{G}_k])^{1/2}\rangle$ as follows:

$$\langle \mathbf{I}, \mathrm{proj}_{\mathcal{H}}(\mathbb{E}[\mathbf{G}_k])^{1/2}\rangle \overset{(a)}{\leq} \sqrt{\|\mathbf{L}\|_{\mathrm{tr}}\langle \mathbf{L}^{-1}, \mathrm{proj}_{\mathcal{H}}(\mathbb{E}[\mathbf{G}_k])\rangle}$$

$$\overset{(b)}{=} \sqrt{\|\mathbf{L}\|_{\mathrm{tr}}\mathbb{E}[\langle \mathbf{L}^{-1}, \mathbf{G}_k\rangle]}$$

$$\overset{(c)}{\leq} \sqrt{\|\mathbf{L}\|_{\mathrm{tr}}\sum_{i=0}^k \mathbb{E}[\|\nabla f(x_i)\|_{\mathbf{L}^{-1}}^2]}$$

$$\overset{(d)}{\leq} \|\mathbf{L}\|_{\mathrm{tr}}^{\frac{1}{1+\nu}} \sqrt{\sum_{i=0}^k \mathbb{E}\left[[f(x_i) - f(x^*)]^{\frac{2\nu}{1+\nu}}\right]}$$

$$\overset{(e)}{\leq} \|\mathbf{L}\|_{\mathrm{tr}}^{\frac{1}{1+\nu}} \sqrt{\sum_{i=0}^k [\mathbb{E}[f(x_i) - f(x^*)]]^{\frac{2\nu}{1+\nu}}}$$

$$\overset{(f)}{\leq} \|\mathbf{L}\|_{\mathrm{tr}}^{\frac{1}{1+\nu}} \sqrt{(k+1)^{\frac{1-\nu}{1+\nu}}\left[\sum_{i=0}^k \mathbb{E}[f(x_i) - f(x^*)]\right]^{\frac{2\nu}{1+\nu}}}$$

$$= \sqrt{k+1}^{\frac{1-\nu}{1+\nu}}\|\mathbf{L}\|_{\mathrm{tr}}^{\frac{1}{1+\nu}}\left[\sum_{i=0}^k \mathbb{E}[f(x_i) - f(x^*)]\right]^{\frac{\nu}{1+\nu}},$$

where (a) uses steps similar to the above calculations; (b) uses the linearity of the expectation and the fact that $\mathbf{L}^{-1} \in \mathcal{H}$, which is implied by Property A1.2 and Assumption 2; (c) uses the definition of $\mathbf{G}_k$; (d) uses Lemma 3; (e) and (f) use the concavity of the function $t \mapsto t^{\frac{2\nu}{1+\nu}}$ for $\nu \in [0, 1]$. $\quad\square$

### F.3  PROOF OF THEOREM 1

Using Lemmas 5 and 6, we get the following inequality:

$$\sum_{k=0}^K \mathbb{E}[f(x_k) - f(x^*)] \leq \tfrac{3}{2}\sqrt{K+1}^{\frac{1-\nu}{1+\nu}}\mathcal{R}\|\mathbf{L}\|_{\mathrm{tr}}^{\frac{1}{1+\nu}}\left[\sum_{k=0}^K \mathbb{E}[f(x_k) - f(x^*)]\right]^{\frac{\nu}{1+\nu}}$$

$$+ \tfrac{3}{2}\sqrt{K+1}\mathcal{R}\|\mathbf{\Sigma}\|_{\mathrm{tr}} + \tfrac{3}{2}\sqrt{\delta}\mathcal{R}\dim(\mathcal{X}),$$

which implies the following inequality:

$$\sum_{k=0}^{K}\mathbb{E}[f(x_k) - f(x^*)] \leq 3\sqrt{K+1}^{1-\nu}\|\mathbf{L}\|_{\mathrm{tr}}\mathcal{R}^{1+\nu}$$
$$+ 3\sqrt{K+1}\|\mathbf{\Sigma}\|_{\mathrm{tr}}\mathcal{R} + 3\sqrt{\delta}\mathcal{R}\dim(\mathcal{X}).$$

It remains to use the convexity property in Assumption 2 and the definition of $\overline{x}_K$ on line 7 of Algorithm 1. $\qquad\square$

# G  PROOFS FOR SECTION 4

## G.1  PROOF OF LEMMA 7

We can upper-bound $\sum_{k=0}^{K}\mathbb{E}[f_k(x^*) - f_k(x_{k+1})]$ as follows:

$$
\begin{aligned}
\sum_{k=0}^{K}\mathbb{E}[f_k(x^*) &- f_k(x_{k+1})] \\
&\overset{(a)}{=} \sum_{k=0}^{K}\alpha_k^{-2}\mathbb{E}[f(\alpha_k x^* + (1-\alpha_k)\overline{x}_k) - f(\alpha_k x_{k+1} + (1-\alpha_k)\overline{x}_k)] \\
&\overset{(b)}{\leq} \sum_{k=0}^{K}\alpha_k^{-2}\mathbb{E}[\alpha_k f(x^*) + (1-\alpha_k)f(\overline{x}_k) - f(\overline{x}_{k+1})] \\
&= \alpha_K^{-2}\mathbb{E}[f(x^*) - f(x_{K+1})] + \alpha_0^{-2}(1-\alpha_0)\mathbb{E}[f(x^*) - f(\overline{x}_0)] \\
&\quad + \sum_{k=1}^{K}(\alpha_k^{-2}(1-\alpha_k) - \alpha_{k-1}^{-2})\mathbb{E}[f(\overline{x}_k) - f(x^*)] \\
&\overset{(c)}{\leq} \tfrac{1}{4}(K+2)^2\mathbb{E}[f(x^*) - f(x_{K+1})],
\end{aligned}
$$

where (a) uses the definition of the functions $f_k(x)$ in eq. (19); (b) uses the definition of $\overline{x}_{k+1}$ on line 7 of Algorithm 2 and the convexity property in Assumption 2; (c) uses the definition $\alpha_k = 2/(k+2)$. $\qquad\square$

## G.2  PROOF OF LEMMA 8

Let $\mathbf{B} = \mathbf{L}$ (the case $\mathbf{B} = \mathbf{\Sigma}$ is analogous). Let $\mathcal{A}_k(\mathbf{Q})\colon \mathbb{S}_{++} \to \mathbb{R}$ be the objective function in eq. (21):

$$
\mathcal{A}_k(\mathbf{Q}) = \langle \mathbf{Q}, \mathbf{L}\mathbf{S}_k \rangle + \langle \mathbf{L}, \delta\mathbf{Q} - \eta^2 \ln(\mathbf{Q}) \rangle. \tag{45}
$$

From Property A1.2, it follows that $\mathbf{H}_k^2 \in \mathcal{H} \cap \mathbb{S}_{++}$. In addition, from the Löwner-Heinz Theorem (Carlen, 2010, Theorem 2.6), it follows that the function $\mathcal{A}_k(\mathbf{Q})$ is convex. Hence, it remains to prove that the first-order stationarity condition holds, that is, the differential of $\mathcal{A}_k(\mathbf{Q})$ is zero on $\mathcal{H}$ at $\mathbf{H}_k^2$:

$$
\mathrm{d}\mathcal{A}_k(\mathbf{H}_k^2)[\mathbf{H}] = 0 \ \text{ for all } \ \mathbf{H} \in \mathcal{H}. \tag{46}
$$

The following Lemma 11 will be used to compute the differential $\mathrm{d}\mathcal{A}_k(\mathbf{Q})[\mathbf{H}]$.

**Lemma 11** ($\downarrow$). *Under Assumption 4, let the function $\mathcal{B}(\mathbf{Q})\colon \mathbb{S}_{++} \to \mathbb{R}$ be defined as follows:*

$$
\mathcal{B}(\mathbf{Q}) = \langle \mathbf{L}, \ln(\mathbf{Q}) \rangle. \tag{47}
$$

*Then the differential of the function $\mathcal{B}(\mathbf{Q})$ for all $\mathbf{Q} \in \mathcal{H} \cap \mathbb{S}_{++}$ is given as follows:*

$$
\mathrm{d}\mathcal{B}(\mathbf{Q})[\mathbf{H}] = \langle \mathbf{L}\mathbf{Q}^{-1}, \mathbf{H} \rangle \ \text{ for all } \ \mathbf{H} \in \mathcal{H}. \tag{48}
$$

Using Lemma 11, we can compute the differential $\mathrm{d}\mathcal{A}_k(\mathbf{H}_k^2)[\mathbf{H}]$ for $\mathbf{H} \in \mathcal{H}$ as follows:

$$
\begin{aligned}
\mathrm{d}\mathcal{A}_k(\mathbf{H}_k^2)[\mathbf{H}] &\overset{(a)}{=} \langle \mathbf{L}(\mathbf{S}_k + \delta\mathbf{I} - \eta^2\mathbf{H}_k^{-2}), \mathbf{H} \rangle \\
&\overset{(b)}{=} \langle \mathbf{L}(\mathbf{S}_k - \mathrm{proj}_{\mathcal{H}}(\mathbf{S}_k)), \mathbf{H} \rangle \\
&\overset{(c)}{=} \langle (\mathbf{S}_k - \mathrm{proj}_{\mathcal{H}}(\mathbf{S}_k)), \mathbf{L}\mathbf{H} \rangle \\
&\overset{(d)}{=} \langle (\mathbf{S}_k - \mathrm{proj}_{\mathcal{H}}(\mathbf{S}_k)), \mathbf{L}\mathbf{H} \rangle
\end{aligned}
$$

where (a) uses Lemma 11; (b) uses eq. (8); (c) uses Assumption 4; (d) uses the fact that $\mathbf{L}\mathbf{H} \in \mathcal{H}$, which is implied by the following Lemma 12.

**Lemma 12** ($\downarrow$). *Under Assumption 4, $\mathbf{L}\mathbf{H} \in \mathcal{H}$ for all $\mathbf{H} \in \mathcal{H}$.*

The proof is now complete. $\qquad\square$

## G.2.1  PROOF OF LEMMA 11

Let constants $a, b \in \mathbb{R}$ be chosen to satisfy the following inequalities:

$$
\mathbf{O} \prec a\mathbf{I} \prec \mathbf{Q} \prec b\mathbf{I}. \tag{49}
$$

Let $\mathbf{H} \in \mathcal{H}$ such that $\|\mathbf{H} - \mathbf{Q}\|_{\mathrm{op}} \leq \min\{(\lambda_{\min}(\mathbf{Q}) - a), b - \lambda_{\max}(\mathbf{Q})\}$. Hence, it is easy to verify that the following inequalities hold:

$$a\mathbf{I} \preceq \mathbf{Q} + \mathbf{H} \preceq b\mathbf{I}. \tag{50}$$

Next, we fix an arbitrary $\epsilon > 0$. By the Weierstrass approximation theorem, there exists a polynomial $p_n(t) = \sum_{i=0}^{n} c_i t^i$ such that $p_n(a) = \ln(a)$, $p_n'(a) = 1/a$, and whose second derivative approximates the function $t \mapsto -1/t^2$ on the segment $[a, b]$ up to the precision $\epsilon$:

$$|p_n''(t) + 1/t^2| \leq \epsilon \quad \text{for all} \ \ t \in [a, b]. \tag{51}$$

From this, using the standard integration arguments, we can conclude that the following approximation inequalities hold for all $t \in [a, b]$:

$$|p_n'(t) - 1/t| \leq \epsilon(b - a), \qquad |p_n(t) - \ln(t)| \leq \tfrac{1}{2}\epsilon(b - a)^2. \tag{52}$$

Further, we obtain the following:

$$|\mathcal{B}(\mathbf{Q} + \mathbf{H}) - \mathcal{B}(\mathbf{Q}) - \int_0^1 \langle \mathbf{L}(\mathbf{Q} + \tau\mathbf{H})^{-1}, \mathbf{H} \rangle \mathrm{d}\tau|$$

$$\overset{(a)}{\leq} |\langle \mathbf{L}, p_n(\mathbf{Q} + \mathbf{H}) - p_n(\mathbf{Q}) \rangle - \int_0^1 \langle \mathbf{L}p_n'(\mathbf{Q} + \tau\mathbf{H}), \mathbf{H} \rangle \mathrm{d}\tau|$$

$$+ \|\mathbf{L}\|_{\mathrm{tr}} \cdot \left(\tfrac{1}{2}\epsilon(b - a)^2 + \tfrac{1}{2}\epsilon(b - a)^2\right) + \|\mathbf{L}\mathbf{H}\|_{\mathrm{tr}} \cdot \epsilon(b - a)$$

$$= |\langle \mathbf{L}, p_n(\mathbf{Q} + \mathbf{H}) - p_n(\mathbf{Q}) \rangle - \int_0^1 \langle \mathbf{L}p_n'(\mathbf{Q} + \tau\mathbf{H}), \mathbf{H} \rangle \mathrm{d}\tau| + \epsilon\left(b^2\|\mathbf{L}\|_{\mathrm{tr}} + b\|\mathbf{L}\mathbf{H}\|_{\mathrm{tr}}\right)$$

$$\overset{(b)}{=} \epsilon\left(b^2\|\mathbf{L}\|_{\mathrm{tr}} + b\|\mathbf{L}\mathbf{H}\|_{\mathrm{tr}}\right).$$

where (a) uses Definition 1, the approximation inequalities above, and the Hölder's inequality for Schatten norms; (b) Uses the fact that $p_n(t)$ is a polynomial and the fact that $\mathbf{Q}\mathbf{L} = \mathbf{L}\mathbf{Q}$ and $\mathbf{H}\mathbf{L} = \mathbf{L}\mathbf{H}$ due to Assumption 4. Next, we take the limit $\epsilon \to 0$ and use the fundamental theorem of calculus and the continuity of the map $\mathbf{Q} \mapsto \mathbf{Q}^{-1}$ on $\mathbb{S}_{++}$, which implies the following:

$$\tfrac{\mathrm{d}}{\mathrm{d}\tau}\mathcal{B}(\mathbf{Q} + \tau\mathbf{H})|_{\tau=0} = \langle \mathbf{L}\mathbf{Q}^{-1}, \mathbf{H} \rangle. \tag{53}$$

Since the right-hand side is continuous in $\mathbf{Q}$, we can conclude that the function $\mathcal{B}(\mathbf{Q})$ is differentiable and its differential is equal to the right-hand side.

### G.2.2 PROOF OF LEMMA 12

Since the operators $\mathbf{L}$ and $\mathbf{H}$ are self-adjoint and commute, they are simultaneously diagonalizeable:

$$\mathbf{L} = \sum_i \lambda_i \cdot u_i \langle u_i, \cdot \rangle \quad \text{and} \quad \mathbf{H} = \sum_i \mu_i \cdot u_i \langle u_i, \cdot \rangle,$$

where $\lambda_i$ and $\mu_i$ are the (possibly repeating) eigenvalues of the operators $\mathbf{L}$ and $\mathbf{H}$, respectively, $\{u_i\} \subset \mathcal{X}$ is an orthonormal basis of the common eigenvectors in the space $\mathcal{X}$. Hence, the operator $\mathbf{L}\mathbf{H}$ is also diagonalizeable as follows:

$$\mathbf{L}\mathbf{H} = \sum_i \lambda_i \mu_i \cdot u_i \langle u_i, \cdot \rangle.$$

Further, let $I_\lambda = \{i : \lambda_i = \lambda\}$ and $J_\mu = \{j : \mu_j = \mu\}$ for arbitrary $\lambda, \mu \in \mathbb{R}$. Let $p(t)$ be a polynomial such that $p(\lambda) = 1$ and $p(\lambda_i) = 0$ for $i \notin I$. Using Property A1.2, we can conclude that

$$p(\mathbf{L}) = \sum_{i \in I_\lambda} u_i \langle u_i, \cdot \rangle \in \mathcal{H}.$$

Similarly, by constructing a polynomial $q(t)$ such that $q(\mu) = 1$ and $q(\mu_j) = 0$ for $j \notin J$ and using Property A1.2, we can show that the following inclusion holds:

$$q(\mathbf{H}) = \sum_{j \in J_\mu} u_j \langle u_j, \cdot \rangle \in \mathcal{H}.$$

Hence, using Property A1.2, we obtain the following inclusion:

$$p(\mathbf{L}) + q(\mathbf{H}) = \sum_{i \in I_\lambda \triangle J_\mu} u_i \langle u_i, \cdot \rangle + 2\sum_{i \in I_\lambda \cap J_\mu} u_i \langle u_i, \cdot \rangle.$$

Finally, we can construct a polynomial $s(t)$ such $s(2) = 1$ and $s(1) = 0$. Using Property A1.2, we can show that

$$s(p(\mathbf{L}) + q(\mathbf{H})) = \sum_{i \in I_\lambda \cap J_\mu} u_i \langle u_i, \cdot \rangle \in \mathcal{H}.$$

From this fact and the above eigendecomposition of the operator $\mathbf{L}\mathbf{H}$, it follows that $\mathbf{L}\mathbf{H} \in \mathcal{H}$. $\quad\square$

### G.3 PROOF OF LEMMA 9

Let $\mathbf{B} = \mathbf{L}$ (the case $\mathbf{B} = \mathbf{\Sigma}$ is analogous). Let functions $l_{-1}(\mathbf{Q}), \ldots, l_k(\mathbf{Q}) \colon \mathbb{S}_{++} \cap \mathcal{H} \to \mathbb{R}$ be defined as follows:

$$l_{-1}(\mathbf{Q}) = \langle \mathbf{L}, \delta \mathbf{Q} - \eta^2 \ln(\mathbf{Q}) \rangle, \quad l_i(\mathbf{Q}) = \|g_i\|_{\mathbf{QL}}^2 \text{ for } i = 0, \ldots, k. \tag{54}$$

Let the operators $\mathbf{Q}_{-1}, \ldots, \mathbf{Q}_k \in \mathcal{H} \cap \mathbb{S}_{++}$ be defined as follows:

$$\mathbf{Q}_{-1} = (\eta^2/\delta)\mathbf{I}, \quad \mathbf{Q}_i = \mathbf{H}_i^2 \text{ for } i = 0, \ldots, k. \tag{55}$$

Using Lemma 8, we can show that the following relation holds for all $i = -1, \ldots, k$:

$$\mathbf{Q}_i = \underset{\mathbf{Q} \in \mathcal{H} \cap \mathbb{S}_{++}}{\arg\min} \sum_{i=-1}^{k} l_i(\mathbf{Q}).$$

Next, we get the following inequality:

$$
\begin{aligned}
\sum_{i=0}^{k} \|g_i\|_{\mathbf{LH}_i^2}^2 &\overset{(a)}{=} \sum_{i=0}^{k} l_i(\mathbf{Q}_i) \\
&= \sum_{i=-1}^{k} l_i(\mathbf{Q}_i) - l_{-1}(\mathbf{Q}_{-1}) \\
&\overset{(b)}{\leq} \sum_{i=-1}^{k} l_i(\mathbf{Q}_k) - l_{-1}(\mathbf{Q}_{-1}) \\
&\overset{(c)}{=} \langle \mathbf{LH}_k^2, \delta\mathbf{I} + \mathbf{S}_k \rangle - \eta^2 \langle \mathbf{L}, \ln(\mathbf{H}_k^2) \rangle - \eta^2 \langle \mathbf{L}, \mathbf{I} \rangle + \eta^2 \langle \mathbf{L}, \ln((\eta^2/\delta)\mathbf{I}) \rangle \\
&\overset{(d)}{=} \langle \mathbf{LH}_k^2, \delta\mathbf{I} + \mathrm{proj}_{\mathcal{H}}(\mathbf{S}_k) \rangle - \eta^2 \langle \mathbf{L}, \ln(\mathbf{H}_k^2) \rangle - \eta^2 \|\mathbf{L}\|_{\mathrm{tr}} + \eta^2 \langle \mathbf{L}, \ln(\eta^2\mathbf{I}/\delta) \rangle \\
&\overset{(e)}{=} \eta^2 \|\mathbf{L}\|_{\mathrm{tr}} - \eta^2 \langle \mathbf{L}, \ln(\delta\mathbf{H}_k^2/\eta^2) \rangle - \eta^2 \|\mathbf{L}\|_{\mathrm{tr}} \\
&\overset{(f)}{=} \eta^2 \langle \mathbf{L}, \ln(\delta\mathbf{I} + \mathrm{proj}_{\mathcal{H}}(\mathbf{S}_k)) \rangle + \eta^2 \|\mathbf{L}\|_{\mathrm{tr}} \ln \tfrac{1}{\delta} \\
&\overset{(g)}{\leq} \eta^2 \|\mathbf{L}\|_{\mathrm{tr}} \ln(\|\delta\mathbf{I} + \mathrm{proj}_{\mathcal{H}}(\mathbf{S}_k)\|_{\mathrm{op}}) + \eta^2 \|\mathbf{L}\|_{\mathrm{tr}} \ln \tfrac{1}{\delta} \\
&\overset{(h)}{\leq} \eta^2 \|\mathbf{L}\|_{\mathrm{tr}} \ln(\|(\delta\mathbf{I} + \mathrm{proj}_{\mathcal{H}}(\mathbf{S}_k))^{1/2}\|_{\mathrm{tr}}^2) + \eta^2 \|\mathbf{L}\|_{\mathrm{tr}} \ln \tfrac{1}{\delta} \\
&\overset{(i)}{=} \eta^2 \|\mathbf{L}\|_{\mathrm{tr}} \ln \left( \tfrac{1}{\delta} \eta^2 \|\mathbf{H}_k^{-1}\|_{\mathrm{tr}}^2 \right),
\end{aligned}
$$

where (a) and (c) use the definition of the functions $l_i(\mathbf{H})$, the definition of the operators $\mathbf{Q}_i$; (b) uses eq. (FTL-BTL); (d) uses Lemma 12, Property A1.2, and the properties of the projection onto $\mathcal{H}$; (e) and (f) use eq. (8) and Definition 1; (g) uses the Hölder's inequality for Schatten norms; (h) uses the inequality $\|\cdot\|_{\mathrm{op}} \leq \|\cdot\|_{\mathrm{tr}}$; (i) uses eq. (8). It remains to take the expectation and use the concavity of the function $t \mapsto \ln(t^2)$ and the Jensen's inequality. $\quad\square$

### G.4 PROOF OF LEMMA 10

Let $n_k = g_k - \nabla f_k(x_k)$ and $r_k = x_k - x^*$. We can obtain the following inequality:

$$
\begin{aligned}
\mathbb{E}[f_k(x_{k+1})] &\overset{(a)}{\leq} \mathbb{E}\Big[ f_k(x_k) + \langle \nabla f_k(x_k), x_{k+1} - x_k \rangle + \tfrac{1}{1+\nu} \alpha_k^{\nu-1} \|\mathbf{L}\|_{\mathrm{tr}}^{\frac{1-\nu}{2}} \|x_{k+1} - x_k\|_{\mathbf{L}}^{1+\nu} \Big] \\
&\overset{(b)}{=} \mathbb{E}\Big[ f_k(x_k) - \langle \nabla f_k(x_k), \mathbf{H}_k g_k \rangle + \tfrac{1}{1+\nu} \alpha_k^{\nu-1} \|\mathbf{L}\|_{\mathrm{tr}}^{\frac{1-\nu}{2}} \|g_k\|_{\mathbf{LH}_k^2}^{1+\nu} \Big] \\
&\overset{(c)}{=} \mathbb{E}\Big[ f_k(x_k) - \|g_k\|_{\mathbf{H}_k}^2 + \langle n_k, \mathbf{H}_k g_k \rangle + \tfrac{1}{1+\nu} \alpha_k^{\nu-1} \|\mathbf{L}\|_{\mathrm{tr}}^{\frac{1-\nu}{2}} \|g_k\|_{\mathbf{LH}_k^2}^{1+\nu} \Big]
\end{aligned}
$$

where (a) uses the definition of the function $f_k(x)$ in eq. (19) and Assumption 2; (b) uses eq. (3) and Assumption 4; (c) uses the definition of $n_k$. Next, similar to the proof of Lemma 5, we can obtain the following inequality:

$$\mathbb{E}[\textstyle\sum_{k=0}^{K} \langle g_k, r_k \rangle] \leq \mathbb{E}[\tfrac{1}{2} \textstyle\sum_{k=0}^{K} \|g_k\|_{\mathbf{H}_k}^2 + \tfrac{1}{2}\mathcal{R}^2 \langle \mathbf{I}, \mathbf{H}_K^{-1} \rangle], \tag{56}$$

Combining this with the previous inequality gives the following:

$$\textstyle\sum_{k=0}^{K} \mathbb{E}[f_k(x_{k+1}) - f_k(x^*)]$$

$$\leq \tfrac{1}{2}\mathcal{R}^2\langle \mathbf{I}, \mathbb{E}[\mathbf{H}_K^{-1}]\rangle + \sum_{k=0}^K \mathbb{E}\left[\langle n_k, \mathbf{H}_k g_k\rangle - \tfrac{1}{2}\|g_k\|_{\mathbf{H}_k}^2 + \tfrac{1}{1+\nu}\alpha_k^{\nu-1}\|\mathbf{L}\|_{\mathrm{tr}}^{\frac{1-\nu}{2}}\|g_k\|_{\mathbf{LH}_k^2}^{1+\nu}\right]$$

$$\overset{(a)}{\leq} \tfrac{1}{2}\mathcal{R}^2\langle \mathbf{I}, \mathbb{E}[\mathbf{H}_K^{-1}]\rangle + \sum_{k=0}^K \mathbb{E}\left[-\tfrac{1}{2}\|g_k\|_{\mathbf{H}_k}^2 + \tfrac{1}{1+\nu}\alpha_k^{\nu-1}\|\mathbf{L}\|_{\mathrm{tr}}^{\frac{1-\nu}{2}}\|g_k\|_{\mathbf{LH}_k^2}^{1+\nu}\right]$$
$$+ \sum_{k=0}^K \mathbb{E}\left[\tfrac{c}{2}\|g_k\|_{\mathbf{\Sigma H}_k^2}^2 + \tfrac{1}{2c}\|n_k\|_{\mathbf{\Sigma}^{-1}}^2\right]$$

$$\overset{(b)}{=} \tfrac{1}{2}\mathcal{R}^2\langle \mathbf{I}, \mathbb{E}[\mathbf{H}_K^{-1}]\rangle + \sum_{k=0}^K \mathbb{E}\left[\tfrac{1}{2}\langle \mathbf{S}_{k-1} - \mathbf{S}_k, \mathbf{H}_k\rangle + \tfrac{1}{1+\nu}\alpha_k^{\nu-1}\|\mathbf{L}\|_{\mathrm{tr}}^{\frac{1-\nu}{2}}\|g_k\|_{\mathbf{LH}_k^2}^{1+\nu}\right]$$
$$+ \sum_{k=0}^K \mathbb{E}\left[\tfrac{c}{2}\|g_k\|_{\mathbf{\Sigma H}_k^2}^2 + \tfrac{1}{2c}\|n_k\|_{\mathbf{\Sigma}^{-1}}^2\right]$$

$$\overset{(c)}{\leq} \tfrac{1}{2}\mathcal{R}^2\langle \mathbf{I}, \mathbb{E}[\mathbf{H}_K^{-1}]\rangle + \tfrac{1}{2}\sum_{k=0}^K \mathbb{E}[\langle \mathbf{S}_{k-1}, \mathbf{H}_{k-1}\rangle - \langle \mathbf{S}_k, \mathbf{H}_k\rangle]$$
$$+ \sum_{k=0}^K \mathbb{E}\left[\tfrac{1}{1+\nu}\alpha_k^{\nu-1}\|\mathbf{L}\|_{\mathrm{tr}}^{\frac{1-\nu}{2}}\|g_k\|_{\mathbf{LH}_k^2}^{1+\nu} + \tfrac{c}{2}\|g_k\|_{\mathbf{\Sigma H}_k^2}^2 + \tfrac{1}{2c}\|n_k\|_{\mathbf{\Sigma}^{-1}}^2\right]$$

$$\overset{(d)}{=} \tfrac{1}{2}(\mathcal{R}^2 - \eta^2)\langle \mathbf{I}, \mathbb{E}[\mathbf{H}_K^{-1}]\rangle + \tfrac{1}{2}\sqrt{\delta}\eta\|\mathbf{I}\|_{\mathrm{tr}}$$
$$+ \sum_{k=0}^K \mathbb{E}\left[\tfrac{1}{1+\nu}\alpha_k^{\nu-1}\|\mathbf{L}\|_{\mathrm{tr}}^{\frac{1-\nu}{2}}\|g_k\|_{\mathbf{LH}_k^2}^{1+\nu} + \tfrac{c}{2}\|g_k\|_{\mathbf{\Sigma H}_k^2}^2 + \tfrac{1}{2c}\|n_k\|_{\mathbf{\Sigma}^{-1}}^2\right]$$

$$\overset{(e)}{\leq} \tfrac{1}{2}(\mathcal{R}^2 - \eta^2)\langle \mathbf{I}, \mathbb{E}[\mathbf{H}_K^{-1}]\rangle + \tfrac{1}{2}\sqrt{\delta}\eta\|\mathbf{I}\|_{\mathrm{tr}} + \tfrac{1}{2c}\|\mathbf{\Sigma}\|_{\mathrm{tr}}\sum_{k=0}^K (1/\alpha_k^2)$$
$$+ \sum_{k=0}^K \mathbb{E}\left[\tfrac{1}{1+\nu}\alpha_k^{\nu-1}\|\mathbf{L}\|_{\mathrm{tr}}^{\frac{1-\nu}{2}}\|g_k\|_{\mathbf{LH}_k^2}^{1+\nu} + \tfrac{c}{2}\|g_k\|_{\mathbf{\Sigma H}_k^2}^2\right]$$

$$\overset{(f)}{\leq} \tfrac{1}{2}(\mathcal{R}^2 - \eta^2)\langle \mathbf{I}, \mathbb{E}[\mathbf{H}_K^{-1}]\rangle + \tfrac{1}{2}\sqrt{\delta}\eta\|\mathbf{I}\|_{\mathrm{tr}} + \tfrac{1}{2c}\|\mathbf{\Sigma}\|_{\mathrm{tr}}\sum_{k=0}^K (1/\alpha_k^2)$$
$$+ \tfrac{c}{2}\mathbb{E}\left[\sum_{k=0}^K \|g_k\|_{\mathbf{\Sigma H}_k^2}^2\right] + \tfrac{1}{1+\nu}\left(\sum_{i=0}^K 1/\alpha_i^2\right)^{\frac{1-\nu}{2}}\|\mathbf{L}\|_{\mathrm{tr}}^{\frac{1-\nu}{2}}\left(\mathbb{E}\left[\sum_{k=0}^K \|g_k\|_{\mathbf{LH}_k^2}^2\right]\right)^{\frac{1+\nu}{2}}$$

$$\overset{(g)}{=} \tfrac{1}{2}(\mathcal{R}^2 - \eta^2)\langle \mathbf{I}, \mathbb{E}[\mathbf{H}_K^{-1}]\rangle + \tfrac{1}{2}\sqrt{\delta}\eta\|\mathbf{I}\|_{\mathrm{tr}} + \left(\sum_{i=0}^K 1/\alpha_i^2\right)^{\frac{1}{2}}\|\mathbf{\Sigma}\|_{\mathrm{tr}}^{\frac{1}{2}}\left(\mathbb{E}\left[\sum_{k=0}^K \|g_k\|_{\mathbf{\Sigma H}_k^2}^2\right]\right)^{\frac{1}{2}}$$
$$+ \tfrac{1}{1+\nu}\left(\sum_{i=0}^K 1/\alpha_i^2\right)^{\frac{1-\nu}{2}}\|\mathbf{L}\|_{\mathrm{tr}}^{\frac{1-\nu}{2}}\left(\mathbb{E}\left[\sum_{k=0}^K \|g_k\|_{\mathbf{LH}_k^2}^2\right]\right)^{\frac{1+\nu}{2}}$$

$$\overset{(h)}{=} \tfrac{1}{2}(\mathcal{R}^2 - \eta^2)\mathbb{E}[\|\mathbf{H}_K^{-1}\|_{\mathrm{tr}}] + \tfrac{1}{2}\sqrt{\delta}\eta\|\mathbf{I}\|_{\mathrm{tr}} + \left(\sum_{i=0}^K 1/\alpha_i^2\right)^{\frac{1}{2}}\|\mathbf{\Sigma}\|_{\mathrm{tr}}^{\frac{1}{2}}\left(\mathbb{E}\left[\sum_{k=0}^K \|g_k\|_{\mathbf{\Sigma H}_k^2}^2\right]\right)^{\frac{1}{2}}$$
$$+ \tfrac{1}{1+\nu}\left(\sum_{i=0}^K 1/\alpha_i^2\right)^{\frac{1-\nu}{2}}\|\mathbf{L}\|_{\mathrm{tr}}^{\frac{1-\nu}{2}}\left(\mathbb{E}\left[\sum_{k=0}^K \|g_k\|_{\mathbf{LH}_k^2}^2\right]\right)^{\frac{1+\nu}{2}},$$

where (a) uses the Young's inequality, Assumption 4, and an arbitrary constant $c > 0$; (b) uses the definition of $\mathbf{S}_k$ in eq. (5); (c) uses eq. (9); (d) uses the definition of $\mathbf{H}_k$ in eq. (8); (e) uses the definition of $n_k$ above, Property A3.2, and the definition of the function $f_k(x)$ in eq. (19); (f) uses the Hölder's inequality, the concavity of the function $t \mapsto t^{\frac{1+\nu}{2}}$, and the Jensen's inequality for the expectation; (g) can be obtained by minimizing in $c > 0$; (h) uses the definition of $\|\cdot\|_{\mathrm{tr}}$. Next, using Lemma 9, we obtain the following technical Lemma 13.

**Lemma 13** ($\downarrow$). *Under the conditions of Lemma 10, for $\mathbf{B} = \mathbf{L}$ or $\mathbf{B} = \mathbf{\Sigma}$, and for all $\gamma \in (0, 1)$, the following inequality holds:*

$$\left(\sum_{i=0}^k 1/\alpha_i^2\right)^{1-\gamma}\|\mathbf{B}\|_{\mathrm{tr}}^{1-\gamma}\left(\mathbb{E}\left[\sum_{i=0}^k \|g_i\|_{\mathbf{BH}_i^2}^2\right]\right)^{\gamma}$$
$$\leq \tfrac{1}{8}\eta^2\mathbb{E}[\|\mathbf{H}_k^{-1}\|_{\mathrm{tr}}] + 2^{\gamma}\left(\sum_{i=0}^k 1/\alpha_i^2\right)^{1-\gamma}\|\mathbf{B}\|_{\mathrm{tr}}\eta^{2\gamma}\ln^{\gamma}(c_k(\mathbf{B}, \gamma)), \tag{57}$$

*where the constant $c(\mathbf{B}, \gamma) > 0$ is defined as follows:*

$$c_k(\mathbf{B}, \gamma) = \max\left\{\exp(1), 2^{3+\gamma}\gamma^{\gamma}\left(\sum_{i=0}^k 1/\alpha_i^2\right)^{1-\gamma}\tfrac{1}{\sqrt{\delta}}\|\mathbf{B}\|_{\mathrm{tr}}\eta^{2\gamma-1}\right\}. \tag{58}$$

Further, using Lemma 13 and the fact that $\|\mathbf{I}\|_{\mathrm{tr}} = \dim(\mathcal{X})$, we obtain the following inequality:

$$\sum_{k=0}^K \mathbb{E}[f_k(x_{k+1}) - f_k(x^*)]$$
$$\leq \left(\tfrac{1}{2}\mathcal{R}^2 - \tfrac{1}{4}\eta^2\right)\mathbb{E}[\|\mathbf{H}_K^{-1}\|_{\mathrm{tr}}] + 2^{\frac{1+\nu}{2}}\left(\sum_{i=0}^K 1/\alpha_i^2\right)^{\frac{1-\nu}{2}}\|\mathbf{L}\|_{\mathrm{tr}}\eta^{1+\nu}\ln\left(c_K\left(\mathbf{L}, \tfrac{1+\nu}{2}\right)\right)$$

$$+ 2^{\frac{1}{2}} \left( \textstyle\sum_{i=0}^{K} 1/\alpha_i^2 \right)^{\frac{1}{2}} \|\mathbf{\Sigma}\|_{\mathrm{tr}} \eta \ln \left( c_K \left( \mathbf{\Sigma}, \tfrac{1}{2} \right) \right) + \tfrac{1}{2} \sqrt{\delta} \eta \dim(\mathcal{X})$$

$$\overset{(a)}{\leq} 2^{\frac{3(1+\nu)}{2}} \left( \textstyle\sum_{i=0}^{K} 1/\alpha_i^2 \right)^{\frac{1-\nu}{2}} \|\mathbf{L}\|_{\mathrm{tr}} \mathcal{R}^{1+\nu} \ln \left( c_K \left( \mathbf{L}, \tfrac{1+\nu}{2} \right) \right)$$

$$+ 2^{\frac{3}{2}} \left( \textstyle\sum_{i=0}^{K} 1/\alpha_i^2 \right)^{\frac{1}{2}} \|\mathbf{\Sigma}\|_{\mathrm{tr}} \mathcal{R} \ln \left( c_K \left( \mathbf{\Sigma}, \tfrac{1}{2} \right) \right) + \sqrt{\delta} \mathcal{R} \dim(\mathcal{X})$$

$$\overset{(b)}{\leq} 2^{\frac{1+5\nu}{2}} \left( \textstyle\sum_{i=1}^{K+2} i^2 \right)^{\frac{1-\nu}{2}} \|\mathbf{L}\|_{\mathrm{tr}} \mathcal{R}^{1+\nu} \ln \left( c_K \left( \mathbf{L}, \tfrac{1+\nu}{2} \right) \right)$$

$$+ 2^{\frac{1}{2}} \left( \textstyle\sum_{i=1}^{K+2} i^2 \right)^{\frac{1}{2}} \|\mathbf{\Sigma}\|_{\mathrm{tr}} \mathcal{R} \ln \left( c_K \left( \mathbf{\Sigma}, \tfrac{1}{2} \right) \right) + \sqrt{\delta} \mathcal{R} \dim(\mathcal{X})$$

$$\overset{(c)}{\leq} 2^{\frac{1+5\nu}{2}} 3^{\frac{\nu-1}{2}} (K+3)^{\frac{3(1-\nu)}{2}} \|\mathbf{L}\|_{\mathrm{tr}} \mathcal{R}^{1+\nu} \ln \left( c_K \left( \mathbf{L}, \tfrac{1+\nu}{2} \right) \right)$$

$$+ 2^{\frac{1}{2}} 3^{-\frac{1}{2}} (K+3)^{\frac{3}{2}} \|\mathbf{\Sigma}\|_{\mathrm{tr}} \mathcal{R} \ln \left( c_K \left( \mathbf{\Sigma}, \tfrac{1}{2} \right) \right) + \sqrt{\delta} \mathcal{R} \dim(\mathcal{X})$$

$$\leq 8 (K+2)^{\frac{3(1-\nu)}{2}} \|\mathbf{L}\|_{\mathrm{tr}} \mathcal{R}^{1+\nu} \ln \left( c_K \left( \mathbf{L}, \tfrac{1+\nu}{2} \right) \right)$$

$$+ 2 (K+2)^{\frac{3}{2}} \|\mathbf{\Sigma}\|_{\mathrm{tr}} \mathcal{R} \ln \left( c_K \left( \mathbf{\Sigma}, \tfrac{1}{2} \right) \right) + \sqrt{\delta} \mathcal{R} \dim(\mathcal{X})$$

where (a) uses the definition $\eta = 2\mathcal{R}$; (b) uses the definition $\alpha_k = 2/(k+2)$; (c) uses the fact that $\sum_{i=1}^{K+2} i^2 \leq \frac{1}{3}(K+3)^3$ and $\nu \leq 1$. Finally, we define $\mathcal{C}_K = 32 \ln \left( \max\{ c_K \left( \mathbf{L}, \tfrac{1+\nu}{2} \right), c_K \left( \mathbf{\Sigma}, \tfrac{1}{2} \right) \} \right)$ and verify that eq. (25) holds. $\qquad\square$

### G.4.1 PROOF OF LEMMA 13

We start with the following inequality:

$$\left( \textstyle\sum_{i=0}^{k} 1/\alpha_i^2 \right)^{1-\gamma} \|\mathbf{B}\|_{\mathrm{tr}}^{1-\gamma} \left( \mathbb{E}\left[ \textstyle\sum_{i=0}^{k} \|g_i\|_{\mathbf{B}\mathbf{H}_i^2}^2 \right] \right)^{\gamma}$$

$$\overset{(a)}{\leq} \left( \textstyle\sum_{i=0}^{k} 1/\alpha_i^2 \right)^{1-\gamma} \|\mathbf{B}\|_{\mathrm{tr}}^{1-\gamma} \left( \eta^2 \|\mathbf{B}\|_{\mathrm{tr}} \ln \left[ \tfrac{1}{\delta} \eta^2 \left( \mathbb{E}[\|\mathbf{H}_k^{-1}\|_{\mathrm{tr}}] \right)^2 \right] \right)^{\gamma}$$

$$\overset{(b)}{=} \left( \textstyle\sum_{i=0}^{k} 1/\alpha_i^2 \right)^{1-\gamma} \|\mathbf{B}\|_{\mathrm{tr}} \left( 2\gamma\eta^2 \ln \left[ \left( \tfrac{\eta}{c\sqrt{\delta}} \right)^{\frac{1}{\gamma}} \left( \mathbb{E}[\|\mathbf{H}_k^{-1}\|_{\mathrm{tr}}] \right)^{\frac{1}{\gamma}} \right] + 2\eta^2 \ln(c) \right)^{\gamma}$$

$$\overset{(c)}{\leq} \left( \textstyle\sum_{i=0}^{k} 1/\alpha_i^2 \right)^{1-\gamma} \|\mathbf{B}\|_{\mathrm{tr}} \left[ \left( 2\gamma\eta^2 \ln \left[ \left( \tfrac{\eta}{c\sqrt{\delta}} \right)^{\frac{1}{\gamma}} \left( \mathbb{E}[\|\mathbf{H}_k^{-1}\|_{\mathrm{tr}}] \right)^{\frac{1}{\gamma}} \right] \right)^{\gamma} + \left( 2\eta^2 \ln(c) \right)^{\gamma} \right]$$

$$\overset{(d)}{\leq} \left( \textstyle\sum_{i=0}^{k} 1/\alpha_i^2 \right)^{1-\gamma} \|\mathbf{B}\|_{\mathrm{tr}} \left[ \left( 2\gamma\eta^2 \right)^{\gamma} \left( \tfrac{\eta}{c\sqrt{\delta}} \right) \mathbb{E}[\|\mathbf{H}_k^{-1}\|_{\mathrm{tr}}] + \left( 2\eta^2 \ln(c) \right)^{\gamma} \right]$$

where (a) uses Lemma 9; (b) uses an arbitrary constant $c > 0$; (c) uses the subadditivity of the function $t \mapsto t^\gamma$; (d) uses the inequality $\ln(t) \leq t$ for $t > 0$. Next, we choose the constant $c > 0$ as follows:

$$c = \max \left\{ \exp(1), \ 2^{3+\gamma} \gamma^\gamma \left( \textstyle\sum_{i=0}^{k} 1/\alpha_i^2 \right)^{1-\gamma} \tfrac{1}{\sqrt{\delta}} \|\mathbf{B}\|_{\mathrm{tr}} \eta^{2\gamma-1} \right\}, \tag{59}$$

which implies the following inequality:

$$\left( \textstyle\sum_{i=0}^{k} 1/\alpha_i^2 \right)^{1-\gamma} \|\mathbf{B}\|_{\mathrm{tr}}^{1-\gamma} \left( \mathbb{E}\left[ \textstyle\sum_{i=0}^{k} \|g_i\|_{\mathbf{B}\mathbf{H}_i^2}^2 \right] \right)^{\gamma}$$

$$\leq \tfrac{1}{8} \eta^2 \mathbb{E}[\|\mathbf{H}_k^{-1}\|_{\mathrm{tr}}] + 2^\gamma \left( \textstyle\sum_{i=0}^{k} 1/\alpha_i^2 \right)^{1-\gamma} \|\mathbf{B}\|_{\mathrm{tr}} \eta^{2\gamma} \ln^\gamma(c)$$

$$\overset{(a)}{\leq} \tfrac{1}{8} \eta^2 \mathbb{E}[\|\mathbf{H}_k^{-1}\|_{\mathrm{tr}}] + 2^\gamma \left( \textstyle\sum_{i=0}^{k} 1/\alpha_i^2 \right)^{1-\gamma} \|\mathbf{B}\|_{\mathrm{tr}} \eta^{2\gamma} \ln(c),$$

where (a) uses the fact that $\ln(c) \geq 1$. $\qquad\square$

