# OpenReview forum: "SGD with Adaptive Preconditioning: Unified Analysis and Momentum Acceleration"
_ICLR.cc/2026/Conference — ICLR 2026 Poster_

### Official Review · Reviewer_JqdH · 2025-10-28

**Soundness:** 3
**Presentation:** 2
**Contribution:** 3
**Rating:** 6
**Confidence:** 4

**Summary:**

This paper presents a unified framework to show the convergence rate of adaptive optimizers under matrix Holder smoothness and more general noise assumptions. They also show the acceleration results of AdaGrad and DASGO with Nesterov momentum.

**Strengths:**

1. The smoothness assumptions and noise assumptions are more general than previous works, making a stronger theoretical contributions.
2. The analysis of combining Nesterov momentum with DASGO is insightful. It helps understand why momentum can benefit Adam in reality, which is not achieved by previous works.
3. The paper tries to connect multiple algorithms including ASGO, DASGO, Muon and Scion.

**Weaknesses:**

1. Assumption 1 seems to be a worse characterization than well-structured preconditioner set in Xie et al. 2025 though they seem to describe the same class of sets. The comparison needs to be done more explicitly. See questions below.
2. In the acceleration part, the additional assumption is pretty restrictive. When the preconditioning is not diagonal, it requires $L$ and $\Sigma$ to be identity matrix, which is a less interesting setting.

**Questions:**

1. It seems that assumption 1 can be shown with the Proposition 3.2 in Xie et al. 2025. Does this suggest that the well-structured preconditioner set is a more fundamental characterization than assumption 1? Can you provide more detailed comparison and insights?

---

### Official Review · Reviewer_YbHY · 2025-10-28

**Soundness:** 2
**Presentation:** 2
**Contribution:** 3
**Rating:** 4
**Confidence:** 4

**Summary:**

The main goal of this paper is to provide a unified analysis of stochastic gradient descent algorithms based on adaptive preconditioning. The authors aim to develop a theoretical framework that encompasses most existing adaptive preconditioned gradient methods, such as AdaGrad, Shampoo, and ASGO. To this end, under suitable assumptions (namely, convexity of the objective function, matrix Hölder smoothness, and bounded variance) the paper presents a unified convergence proof that recovers known guarantees for many state-of-the-art methods in stochastic optimization. A secondary contribution concerns the study of a Nesterov-type acceleration of the proposed algorithm, showing, for instance, that AdaGrad can benefit simultaneously from both momentum and diagonal preconditioning.

The contribution is interesting; however, the paper is written in a rather technical style and is primarily accessible to experts in the theoretical analysis of adaptive preconditioned stochastic gradient methods. One of the strengths of the work lies in its careful comparison of the proposed results with existing literature. Nevertheless, as I am not a specialist in the most recent developments of SGD with adaptive preconditioning, it is difficult for me to fully assess the novelty of the contribution, particularly with respect to Gupta et al. (2017), who also provided a unified convergence framework for such algorithms, and the recent preprint by Xie et al. (2025), which the authors mention as a concurrent unified analysis of AdaGrad-type methods.

Overall, the paper has merit, and its contribution is relevant to the community. However, I believe the manuscript requires substantial improvements to better convey the significance and impact of its main results relative to existing work. I would appreciate clarifications and responses from the authors on the key points raised below. While I do not consider the paper ready for publication in its current form, I would be inclined to increase my rating if the exposition were improved and several of the issues outlined below were addressed.

**Strengths:**

* Provides solid theoretical results through a unified analysis of stochastic gradient descent with adaptive preconditioning.
* Offers a clear and well-structured overview of prior work.
* Demonstrates overall rigor in the presentation of assumptions and theoretical arguments.
* Includes a relevant and insightful analysis of the Nesterov acceleration applied to the proposed algorithm.

**Weaknesses:**

* The level of novelty of the paper is difficult to assess in comparison with existing work.
* The paper lacks numerical experiments that could illustrate the potential benefits of the theoretical results and provide insights into the empirical performance of algorithms encompassed by the proposed unified analysis.

**Questions:**

* In Section 2.1, Definition 1 and Assumption 1 establish the framework for the unified preconditioning setting. Could the authors comment on the novelty of this definition and assumption compared with previous works that also proposed unified analyses of SGD algorithms with adaptive preconditioning ?

* What distinguishes Lemma 1 from similar results in Gupta et al. (2017)?

* The authors claim that Lemma 2 highlights a key difference from Gupta et al. (2017). Could they elaborate further on this point ?

* At the end of page 4 and in Appendix B, it is mentioned that the unified analysis of Xie et al. (2025) shares many similarities with the results presented in Sections 2 and 3, and that the main distinction lies in Section 4, since Xie et al. (2025) do not include results involving momentum acceleration. I therefore wonder whether the main novelty of this paper essentially lies in Section 4, and if so, whether the authors could place stronger emphasis on this contribution.

* The function $\mathcal{R}(x)$ defined in Equation (12) appears to play a central role in the unified analysis presented in Section 3. Is the use of such a function novel in the analysis of adaptive preconditioned SGD algorithms? How does it connect to existing literature on this topic? It seems that existing convergence proofs for specific algorithms (e.g.\ AdaGrad) may already rely on similar quantities - could the authors clarify this point?

* Could the proposed unified analysis provide deeper insights into the empirical performance of adaptive preconditioned SGD algorithms compared to existing convergence proofs or algorithm-specific theoretical analyses?

---

### Official Review · Reviewer_3EqR · 2025-10-31

**Soundness:** 3
**Presentation:** 3
**Contribution:** 2
**Rating:** 2
**Confidence:** 3

**Summary:**

The paper studies the theoretical aspects of preconditioned optimizers, including ASGO/One-Sided Shampoo, AdaGrad, and DASGO. The paper makes an attempt to unify the analysis of such type of algorithms under specific assumptions, as well as acceleration results of the optimizers with diagonal preconditioners.

**Strengths:**

1. The paper is written in an easy-to-follow way, making all the results and notations easy to understand.
2. The convergence results are obtained under Holder smoothness, which is a more general assumption compared to the existing works for the same algorithms.

**Weaknesses:**

1. There seems to be some overclaims in the contribution part. Firstly, there has been established work on the acceleration of adaptive gradient methods. I kindly refer the authors to [1] for acceleration results of adaptive gradient methods in the diagonal preconditioner case. Also, the convergence of DASGO can actually be covered by [2] in Theorem 3.11 for block-wise RMSProp, making the contribution of this paper kind of not significant enough.
2. Assumption 4 in the acceleration part is very restrictive. It is not satisfied by the majority of algorithms with preconditioners that are not diagonal, making the analysis actually **not** a valid analysis for a general framework.



[1] Ene, Alina, Huy L. Nguyen, and Adrian Vladu. "Adaptive gradient methods for constrained convex optimization and variational inequalities." Proceedings of the AAAI Conference on Artificial Intelligence. Vol. 35. No. 8. 2021.

[2] Xie, Shuo, Mohamad Amin Mohamadi, and Zhiyuan Li. "Adam Exploits $\ell_\infty $-geometry of Loss Landscape via Coordinate-wise Adaptivity." arXiv preprint arXiv:2410.08198 (2024).

**Questions:**

1. In Theorem 1, the authors mentioned an almost surely upper bound $ R $. Will there be an estimation for $ R $? I understand that for both AdaGrad or ASGO/One-Sided Shampoo, the boundedness of $ R $ can be guaranteed by imposing a projection operation. Will this also work in this framework?
2. I think the major difference of the paper compared to [3] is the employment of Holder smoothness. Could the authors provide some discussions on how such imrprovements can help better explain the practicalness of the analyzed algorithms.



[3] Xie, Shuo, et al. "Structured preconditioners in adaptive optimization: A unified analysis." arXiv preprint arXiv:2503.10537 (2025).

---

### Official Review · Reviewer_rtr6 · 2025-11-03

**Soundness:** 4
**Presentation:** 4
**Contribution:** 3
**Rating:** 8
**Confidence:** 2

**Summary:**

This paper presents a unified convergence analysis for a class of adaptive stochastic gradient descent (SGD) methods. The analysis is developed for stochastic convex objectives under matrix Hölder smoothness and bounded variance assumptions. The authors' first contribution is a general convergence proof (Theorem 1) that recovers state-of-the-art rates for many adaptive methods and provides the first theoretical guarantees for DASGO. The second main contribution is an accelerated algorithm (Algorithm 2) that combines Nesterov momentum with diagonal preconditioning (e.g., AdaGrad, DASGO).

**Strengths:**

1. The paper offers a clear and unified theoretical analysis of adaptive optimization methods. Its findings are novel, with a single theorem (Theorem 1) that recovers known bounds for several adaptive algorithms and establishes the first convergence guarantees for DASGO. The framework’s generality extends to both the smoothness and variance assumptions, and the technical contributions are strong.
2. Algorithm 2 enhances adaptive methods with diagonal preconditioning by incorporating momentum, achieving provable convergence with acceleration. This represents both an innovative algorithmic development and a significant analytical advancement.

**Weaknesses:**

1. Several results implicitly require the smoothness and noise operators (e.g., $L,\Sigma$) to live in the same structured space as the preconditioner $\mathcal H$ (e.g., Assumption 2, even before the acceleration results). For diagonal $\mathcal H$, this effectively enforces axis-aligned curvature/noise, leaving other cases out of scope. In problems where principal directions are not coordinate-aligned, the guarantees may not hold.

2. The theory sets $\eta$ proportional to a radius $R \ge R(x^\*)$ (or similar), which is rarely, if ever, specified in deep learning practice. A significance of AdaGrad-style methods is being parameter-free (or nearly so); requiring $R$ undermines that appeal.

3. The accelerated variant uses a time-varying Nesterov schedule (e.g., $\alpha_k = 2/(k+2)$) and gradients of auxiliary $f_k$, which diverges from the standard “constant-momentum” implementations (e.g., $\beta \approx 0.9$ in Adam) used in large-scale training.

4. It would be interesting to explore whether further analysis can be done in non-convex setting (or under the PL condition).

**Questions:**

- How should the parameter $\delta$ be chosen to achieve the optimal guarantee? In practice, it is typically set to a very small value for numerical stability. However, according to Theorem 2, choosing a smaller $\delta$ may actually lead to a larger upper bound due to its presence in the logarithmic term.
- Does corresponding lower bound results exist in this setting (or a similar setting)?

---

### Meta-Review · Area_Chair_4PDy · 2026-01-06

**Summary:**

This paper is a theory submission on stochastic optimization of convex objectives with AdaGrad-type adaptive preconditioning. The authors propose a unified analysis framework that covers both anisotropic and matrix Hölder smoothness (covering smooth, non-smooth, and intermediate regimes). Within this framework, they recover known rates for several well-known methods (for example, AdaGrad-Norm, AdaGrad, and ASGO, also known as One-sided Shampoo), and they also establish a new theoretical guarantee for DASGO, clarifying its relationship to Scion.

The most distinctive part is the momentum section. The paper proposes an accelerated algorithm that applies preconditioned SGD to a time-varying surrogate objective with a Nesterov-style schedule, and proves an accelerated guarantee under an additional commutativity requirement. The central concerns in the reviews primarily focus on significance and positioning rather than correctness: how clearly the acceleration and DASGO guarantees differ from prior accelerated adaptive-gradient work and recent unified analyses, and how much the paper’s narrative about practical optimizers is supported, given the absence of experiments.

Overall, I see this as borderline but leaning towards acceptance. The technical development is substantial, and no reviewer identifies a clear soundness flaw; however, the paper requires clearer positioning of its novelty and at least minimal empirical evidence to fully support its claims of practical relevance. The authors are strongly encouraged to incorporate the reviewers’ suggestions in the final version, since the recommendation to accept is made with the expectation that these changes will be implemented.

**Reviewer Concerns:**

A recurring point in the reviews is how to interpret the smoothness and noise operators being chosen within the same structured family as the preconditioners. In my view, this is best read as a statement about when the bounds meaningfully capture the benefit of the chosen structure: the rates reflect the advantage of diagonal or structured preconditioning only when the smoothness and noise are well represented in that same structure. It would be helpful if the paper stated this more explicitly, as otherwise, readers may interpret it as a stronger limitation than intended.

The main concerns that remain are novelty, clarity, and evidence. Some reviewers argue that the paper’s “first” claims regarding acceleration and DASGO require sharper positioning, particularly in relation to existing accelerated adaptive-gradient methods and recent unified analyses. Separately, the paper lacks experiments, which is a significant weakness given that the motivation and claimed implications are partly practice-facing.

**Reviewer Scores:**

I would expect limited movement in scores unless the authors substantially tighten the novelty positioning and add at least minimal empirical evidence. Reviewer 3EqR is the most skeptical about novelty and overclaiming; if the authors directly address the cited prior acceleration work and narrow the “first” claims appropriately, I could see them moving to 4. Reviewer YbHY could move up to 6 if the authors make the novelty relative to prior unified analyses much clearer and add a small experimental section. The other reviewers are likely to stick with their scores.

---

### Decision · Program_Chairs · 2026-01-26

Accept (Poster)